# DATA-AUGMENTED FEW-SHOT NEURAL EMULATOR FOR COMPUTER-MODEL SYSTEM IDENTIFICATION

## ABSTRACT

Partial differential equations (PDEs) underpin the modeling of many natural and engineered systems. It can be convenient to express such models as neural PDEs rather than using traditional numerical PDE solvers by replacing part or all of the PDE's governing equations with a neural network representation. Neural PDEs are often easier to differentiate, linearize, reduce, or use for uncertainty quantification than the original numerical solver. They are usually trained on solution trajectories obtained by long-horizon rollout of the PDE solver. Here we propose a more sample-efficient data-augmentation strategy for generating neural PDE training data from a computer model by space-filling sampling of local "stencil" states. This approach removes a large degree of spatiotemporal redundancy present in trajectory data and oversamples states that may be rarely visited but help the neural PDE generalize across the state space. We demonstrate that accurate neural PDE stencil operators can be learned from synthetic training data generated by the computational equivalent of 10 timesteps' worth of numerical simulation. Accuracy is further improved if we assume access to a single full-trajectory simulation from the computer model, which is typically available in practice. Across several PDE systems, we show that our data-augmented stencil data yield better trained neural stencil operators, with clear performance gains compared with naïvely sampled stencil data from simulation trajectories. Finally, with only 10 solver steps' worth of augmented stencil data, our approach outperforms traditional ML emulators trained on thousands of trajectories in long-horizon rollout accuracy and stability.

## 1 INTRODUCTION

Mechanistic computer models, often formulated as partial differential equations (PDEs), are pivotal for simulating complex physical systems across fluid dynamics (Kaushik et al., 2015; Li et al., 2023), climate modeling (Wang et al., 2009; McGuffie & Henderson-Sellers, 2001), biology (Cerrolaza et al., 2017), and chemistry (Evans, 2022). These PDE-based models simulate underlying governing processes to predict complex dynamics, informing decision-making, system design, and intervention strategies. In practice, PDEs often lack analytical solutions and rely on classical numerical methods such as finite difference methods (FDM) (LeVeque, 2007; Strikwerda, 2004), finite volume methods (FVM) (Moukalled et al., 2015; Versteeg & Malalasekera, 2007), and finite element methods (FEM) (Logg et al., 2012; Zienkiewicz & Taylor, 2005). These discretization approaches approximate differential operators via handcrafted stencils, balancing simplicity (FDM), local conservation (FVM), and geometric flexibility (FEM). For applications requiring increased accuracy, high-order schemes like discontinuous Galerkin and spectral methods have emerged (Hesthaven & Warburton, 2007; Cockburn et al., 2000), at higher computational cost and complexity. All of these rely on correctly specifying the governing equations, a requirement that breaks down when the underlying physics are only partially known or prohibitively complex (Quarteroni & Valli, 2008).

Emerging machine learning (ML) approaches seek to learn solution operators or surrogate models directly from data while retaining physical fidelity and improving flexibility and scalability (Raissi et al., 2019; Bar-Sinai et al., 2019; Brandstetter et al., 2022). Physics-informed neural networks (PINNs) (Lagaris et al., 1998; Raissi et al., 2019; Cai et al., 2021; Luo et al., 2025) embed PDE residuals into the loss, enabling mesh-free, often unsupervised training. Neural operators–e.g. DeepONets (Lu et al., 2021), Fourier Neural Operators (FNOs) (Li et al., 2021)–learn mappings between infinite-dimensional function spaces, generalizing across inputs without discretization. Despite strong

promise, both approaches often endure slow or unstable convergence, spectral bias, and sensitivity to loss-weight choices (Krishnapriyan et al., 2021). Heuristics such as adaptive sampling (Mao & Meng, 2023; Tang et al., 2024) and weight-update rules (McClenny & Braga-Neto, 2023; Xiang et al., 2022; Wang et al., 2024) mitigate specific failure modes, but broader challenges–especially efficient, few-shot learning of surrogates for forecasting–remain open.

An alternative to regression-style ML surrogates is neural differential equations (NDEs), which learn the governing equations or "right-hand-side" (RHS) of an ordinary (ODE) or partial differential equation with a neural network, such as U-Net (Takamoto et al., 2022), and make predictions by timestepping the learned equations using a numerical solver (Chen et al., 2018; Akhare et al., 2025). When only part of the RHS is learned, the models are termed universal differential equations (UDEs) (Rackauckas et al., 2020) or hybrid models (Melland et al., 2021; Kochkov et al., 2021; Yu et al., 2023). NDEs–including neural ODEs (NODEs) and neural PDEs (NPDEs)–inherit the computational cost of simulation models because they are themselves ODE or PDE solvers, yet they offer advantages over (possibly physics-constrained) regression-type surrogates. Even when governing equations are known, recasting a large, complex simulation as an NPDE can simplify equation-level manipulation–for example, linearizing NPDE for stability analysis (Brenowitz & Bretherton, 2019) or obtaining gradients via adjoint or automatic differentiation–relative to linearizing or differentiating the original codebase. When the true governing equations are not known but are imperfectly approximated by a simulator, an NDE emulator of that simulator can serve as a "prior" over unknown system dynamics that can be updated with measurements (DeGennaro et al., 2019). Moreover, intrusive model-reduction techniques can be applied directly to the NPDE to accelerate simulation while avoiding the software complexity of modifying a mature codebase (Chen, 2012; DeGennaro et al., 2019; Prakash & Zhang, 2025). Some approaches learn resolution-specific local discretizations for stable long-horizon rollouts on coarse grids (Maddu et al., 2023) , whereas autoregressive next-step predictors (Bar-Sinai et al., 2019; Hsieh et al., 2019) iteratively apply a neural network to advance the state–an approach akin to discrete flow-map learning under partial observation (Churchill & Xiu, 2022).

Learning the governing equations of PDE systems in neural representations can be viewed as a special form of system identification, and NPDEs/UDEs are commonly trained on long solution trajectories. When data are generated from a simulation code rather than experiments, we can greatly control the training data. Specifically, in a simulation, we can precisely control the initial and boundary conditions, domain size, grid resolution, timestep, and other numerics, enabling *designing* more sample-efficient training sets for system identification and neural PDE training. Yet PDE solutions exhibit strong spatiotemporal redundancy–neighboring cells and successive timesteps often contain very similar states–so long integrations expend compute on simulating states that provide little independent information about the system's governing dynamics. We instead sample from a space-filling design of local stencil states, emphasizing statistically independent configurations and underrepresented regions of state space.

To achieve this, we observe that numerical PDE solvers are often implemented in terms of a *stencil operator*: a spatially discretized RHS of the PDE determining the evolution of a grid cell's state as a function of the state vector in a local "stencil" neighborhood of that cell (Caramia & Distaso, 2025). Due to locality and homogeneity of the PDE's governing equations, the same stencil operation is applied at every grid cell at every timestep. Rather than collecting an ensemble of costly, high-dimensional solutions, we learn this stencil mapping from large numbers of computationally inexpensive *stencil* evaluations. This corresponds to running the simulator over the neighborhood of a single grid cell for a single timestep across a statistically designed collection of local states.

At the core of our proposed approach lies a **neural stencil emulator (NSE)** that performs system identification in function space: it infers the model structure or functional form of the PDE RHS directly from grid-cell simulation data. NSE training leverages large amounts of grid-cell-level simulation data that are inexpensive to generate and often with no more than a single simulation. This non-intrusive scheme combines the scalability of data-driven models with the interpretability of mechanistic stencils, yielding an efficient, physics-aware surrogate for PDE evolution.

Our contributions are as follows:

- We introduce the *Neural Stencil Emulator* (**NSE**), a non-intrusive, data-driven system-identification framework that learns the governing equations of computer models from inexpensive stencil evaluations, enabling stable forecasting;

- We develop several data-augmentation strategies, including a novel PCA-based scheme, that improves our emulator's sample efficiency and few-shot generalization to unseen initial conditions using only a handful of full-order simulation snapshots;

- We demonstrate NSE's effectiveness across multiple PDE systems, achieving low errors relative to full-order solutions, and show superior long-horizon rollout accuracy and stability compared to widely used baselines (FNO, U-Net, and PINN) under limited simulation budget.

## 2 PRELIMINARIES

In this paper, vectors and matrices are denoted by bold lowercase and uppercase letters, respectively.

### 2.1 DYNAMICAL MODEL

**Continuous formulation.** Let $u(\boldsymbol{x}, t)$ denote a spatio-temporal variable at spatial location $\boldsymbol{x} \in \Omega = [0, L]^d$ and time $t \in [0, T]$. The evolution of $u$ is governed by the nonlinear partial differential equation capturing the system dynamics via a nonlinear operator $\mathcal{F}$ that parameterizes the time derivative $\partial u / \partial t$ and the initial condition $u(\boldsymbol{x}, t_0)$. Formally, such a PDE can be expressed as:

$$\frac{\partial u(\boldsymbol{x}, t)}{\partial t} = \mathcal{F}(u(\boldsymbol{x}, t)) \qquad (\boldsymbol{x}, t) \in \Omega \times [0, T], \tag{1a}$$

$$u(\boldsymbol{x}, t_0) = u_0(\boldsymbol{x}) \qquad \boldsymbol{x} \in \Omega, \tag{1b}$$

$$\mathcal{BC}\big(u(\boldsymbol{x}, t)\big) = 0 \qquad (\boldsymbol{x}, t) \in \partial\Omega \times [0, T], \tag{1c}$$

where $\mathcal{F}$ encodes the system dynamics and $\mathcal{BC}(\cdot)$ enforces the boundary condition applied on $\partial\Omega = \bigcup_{k=1}^{d}\{\boldsymbol{x} \in \mathbb{R}^d \mid 0 \leq x_j \leq L \quad \text{for all } j \neq k, \ x_k \in \{0, L\}\}$. Lastly, the initial condition $u(\boldsymbol{x}, t_0)$ completes the PDE formulation, enabling the solution of $u(\boldsymbol{x}, t)$ over time.

**Space–time discretization.** To solve the aforementioned PDE numerically, one can discretize the spatial domain $\Omega$ on $n$ grid points $\{\boldsymbol{x}_i\}_{i=1}^{n}$ and partition the temporal domain into uniform steps of size $\Delta t$. Setting $t_k = t_0 + k\Delta t$, we can approximate the continuous $u(\boldsymbol{x}, t)$ variable by discretizing the state at $t_k$ by $\boldsymbol{u}^{(k)} = \{u(\boldsymbol{x}_i, t_k)\}_{i=1}^{n}$. Accordingly, a first-order explicit scheme yields

$$\boldsymbol{u}^{(k+1)} = \boldsymbol{u}^{(k)} + \Delta t\, \mathbf{F}\big(\boldsymbol{u}^{(k)}\big), \qquad \mathbf{F} : \mathbb{R}^n \to \mathbb{R}^n, \tag{2}$$

where $\mathbf{F}$ is the discrete counterpart of $\mathcal{F}$.

### 2.2 SYSTEM IDENTIFICATION

In a numerical method, a mechanistic PDE solver advances a discretized PDE state by integrating it in time. As an example, an explicit 2D finite-difference solver using an Euler timestepping can be formulated via a localized *stencil* operator $\mathbf{F}$ at each grid cell $(i, j)$ and time $t$ as

$$u_{i,j}^{t+1} = u_{i,j}^{t} + \mathbf{F}(\boldsymbol{S}(u_{i,j}^{t}))\Delta t$$

Here, $u_{i,j}^{t}$ denotes the state at grid cell $(i, j)$ and time $t$. $\mathbf{S}(\cdot)$ is the localized *stencil* at $u_{i,j}^{t}$: a set containing $u_{i,j}^{t}$ and its neighboring grid cells within the fixed domain of dependence defining the stencil operator. For example, a 5-point stencil at $u_{i,j}^{t}$ is a set $\{u_{i,j}^{t}, u_{i-1,j}^{t}, u_{i+1,j}^{t}, u_{i,j-1}^{t}, u_{i,j+1}^{t}\}$. Such stencils are used to approximate spatial derivatives at each grid cell.

## 3 NEURAL STENCIL EMULATOR

In Equation (1), we ideally wish to recover the continuum PDE's RHS $\mathcal{F}(\cdot)$; but in practice we learn the discretized operator $\mathbf{F}$ and implement it within a numerical solver. While $\mathbf{F}$ is known analytically for simple PDEs, complex simulation codes often contain conditional logic, empirical closures, and other intricate parameterizations that preclude a closed-form stencil operator. This lack of an analytical expression for $\mathbf{F}$ has traditionally limited the application of intrusive reduced-order models (ROMs) that require direct access to the governing equations.

Instead, we propose to learn a statistical representation of $\mathbf{F}$ from PDE solution data, rendering model order reduction non-intrusive. Our approach treats $\mathbf{F}$ as a black-box mapping

$$\boldsymbol{S}(u_{i,j}^t) \mapsto u_{i,j}^{t+1}, \qquad u_{i,j}^{t+1} = u_{i,j}^t + \mathbf{F}\big(\boldsymbol{S}(u_{i,j}^t)\big)\,\Delta t,$$

and trains a machine learning model $\widehat{\mathbf{F}}_{\boldsymbol{\theta}}$ with parameters $\boldsymbol{\theta}$ to approximate it. We emphasize here that each stencil evaluation is low-dimensional (e.g. five or thirteen inputs plus one output), so a single high-fidelity simulation can produce a large number of training examples. For instance, a climate model with $10^6$ grid cells over $10^6$ timesteps yields $\sim 10^{12}$ stencil evaluations. Moreover, compared with existing state-of-the-art PDE surrogates based on PINN or neural operator learning that train on full spatial field instances, the stencil input is orders of magnitude lower dimensional, significantly reducing model size and training cost.

Various ML models can serve as $\widehat{\mathbf{F}}_{\boldsymbol{\theta}}(\cdot)$. One natural choice is Sparse Identification of Nonlinear Dynamics (SINDy) (Brunton et al., 2016), which performs sparse regression on a set of nonlinear functions of state snapshots versus derivatives to identify the governing equations. While effective for "clean" PDE systems, SINDy's basis may be too restrictive for the complex parameterizations in large-scale simulations. Here we instead use a neural-network-based stencil emulator, which is a more flexible and expressive functional approximator, for learning the RHS of high-dimensional PDEs (see Fig. 1), hence the name *neural stencil emulator* (**NSE**). We use a standard mean squared error loss over a dataset $\mathcal{D} = \{\boldsymbol{S}_z(u_z),\, \delta u_z\}_{z=1}^Z$, where $\delta u_z = (u_z^{t+1} - u_z^t)/\Delta t$ and $z$ denotes spatial coordinate:

$$\mathcal{L}(\boldsymbol{\theta}) = \frac{1}{Z}\sum_{z=1}^Z \big\|\widehat{\mathbf{F}}_{\boldsymbol{\theta}}(\boldsymbol{S}_z) - \delta u_z\big\|^2.$$

Optionally, physics-informed penalties (e.g. enforcing $\nabla \cdot u = 0$) can be added.

Once trained, our NSE, $\widehat{\mathbf{F}}_{\boldsymbol{\theta}}$, replaces conventional timestepping, evolving the state while maintaining high accuracy relative to the original solver. By approximating $\mathbf{F}$ directly–rather than the full state evolution–our surrogate remains non-intrusive and readily deployable alongside legacy solvers.

### 3.1 ADAPTIVE SAMPLING STRATEGIES

Stencil data from a single PDE trajectory are highly redundant: neighboring grid cells and successive timesteps are strongly correlated, so the *effective sample size* is far smaller than the total stencil evaluations obtained. To achieve desired sample efficiency in NSE training, we design space-filling and adaptive sampling strategies that (i) decorrelate stencil samples and (ii) increase coverage of rare but dynamically relevant states. By *synthetically* constructing localized stencil states and forcing the simulator to evaluate them, we

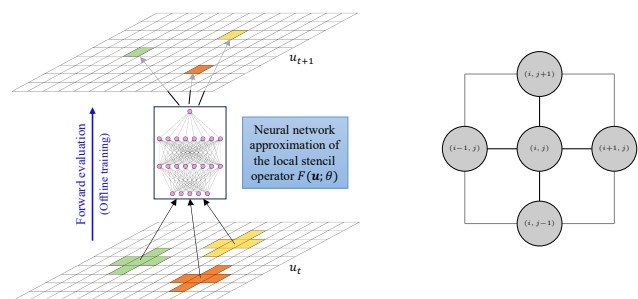

Figure 1: *Left.* A neural network learns the *stencil operator* $\mathbf{F}$: a mapping from a localized stencil of a state ($\boldsymbol{S}$) to its finite difference-based time derivative *Right.* A 5-point stencil schematic.

avoid long integrations and enable few-shot learning. Hence, we use the forward model to obtain one-step labels $(\boldsymbol{S}, \delta u)$ for supervised learning of the stencil operator.

#### 3.1.1 PURE STRATEGIES.

**On-trajectory ("Short-Traj") sampling.** As a baseline, we use all available stencils from a given simulation trajectory when only short simulation is available. This *ergodic* sampler provides dynamically common states but yields strong space–time correlations and under-represents tail events.

**Off-trajectory synthetic sampling.** We introduce two synthetic stencil design and sampling schemes that expand coverage of rare (tail) states in NSE training data, particularly when only short simulation trajectories are available.

**(i) Random sampling.** Sample $\widetilde{\boldsymbol{S}} \in [0,1]^m$ via i.i.d. uniform draws or Sobol' quasi-random sequences for space-filling coverage. Affinely rescale each state within a sampled stencil to the simulation data range. These i.i.d.-like inputs increase effective sample size and oversample corners of state space, promoting interpolation rather than extrapolation at training time for rare states.

**(ii) PCA-guided design.** Given a collection of actual on-trajectory stencils $\{\boldsymbol{S}_z\}$, we compute PCA using stencil mean $\boldsymbol{\mu}$, loading matrix $P \in \mathbb{R}^{m \times r}$, and PC scores $\mathbf{Z} = P^\top (\boldsymbol{S} - \boldsymbol{\mu})$. We set $r = m$ (possible as $Z > m$ in our case) yielding full PCA. Next, we construct a hyper-rectangle in PC space using per-PC minima and maxima $[L_k, U_k]$ observed in $\mathbf{Z}$. Then perform the following steps:

1. Draw $\tilde{\boldsymbol{z}} \in [0,1]^m$ via uniform or Sobol' sampling and map to PC ranges: $\hat{\mathbf{z}}_k = L_k + \tilde{\mathbf{z}}_k(U_k - L_k)$.

2. Back-project to the state space: $\hat{\boldsymbol{S}} = \boldsymbol{\mu} + P\hat{\mathbf{z}}$.

3. Filter out stencil samples outside simulation data range (applying physical constraints).

4. Repeat until the desired number of synthetic stencil states are collected.

This targets high-variance directions while retaining plausibility inherited from the empirical PCs.

### 3.1.2 Mixed strategies.

To balance frequent and rare regimes, we form a mixture sampler combining: (i) stencils from an ergodic simulation and (ii) auxiliary stencils with one-step evaluations obtained using solver.

**On-trajectory downsampling.** A single long full-order simulation can produce millions of near-duplicate stencils. In our mixed strategies, we assume access to one such run and first downsample its stencil states (uniformly in space–time) to reduce correlation (Fig. 2). Next, we assume that we have a small additional compute budget to further augment the downsampled set by evaluating an equal-sized batch obtained via one of four options: (a) a short burst from a new initialization, (b) a short extension from the run's terminal state, (c) one-step evaluations of synthetically designed random stencils, or (d) one-step evaluations of PCA-guided synthetic stencils.

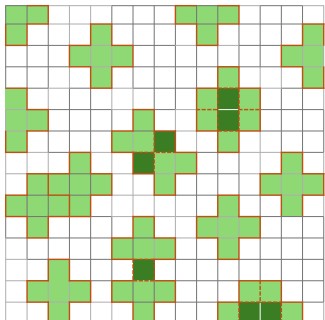

Figure 2: *On-trajectory downsampling.* We downsample stencils uniformly in space-time to increase efficiency in our mixed strategies for training neural stencil emulators.

**(i) Downsampled+Diff Init:** Combine downsampled stencils with short simulation from a *new* initialization to the same stencil count.

**(ii) Downsampled+Extend:** Combine downsampled stencils with a short extension of the original simulation from its terminal state to the same stencil count.

**(iii) Downsampled+Random:** Combine downsampled stencils with an equal number of one-step evaluations of randomly generated stencils via uniform or Sobol' sampling.

**(iv) Downsampled+PCA:** Combine downsampled stencils with an equal number of one-step evaluations of PCA-guided stencils generated with uniform or Sobol' space filling design.

These four data augmentation choices yield four mixed strategies that combine the downsampled on-trajectory data with one of the pure strategies from Section 3.1.1. These mixes reduce redundancy while injecting state-space diversity.

**Remark.** As per-dimension grid size $N$ grows on a $d$-dimensional grid, the stencil size $m$ and NSE $\widehat{\mathbf{F}}_{\boldsymbol{\theta}}$ remain fixed, while the number of available stencils scales as $Z \propto N^d$. With fixed spatial $\Delta x$, doubling the domain in each dimension doubles $N$ and multiplies $Z$ by $2^d$. Consequently, a model trained at lower domain sizes can be rolled out at higher domain sizes provided $\Delta x$, boundary conditions, and the timestep $\Delta t$ are unchanged. This underscores NSE's few-shot aspect.

## 4 Experiments

We test our Neural Stencil Emulator (NSE) on canonical PDEs to evaluate:

1. how sampling design affects data efficiency and forecast accuracy;

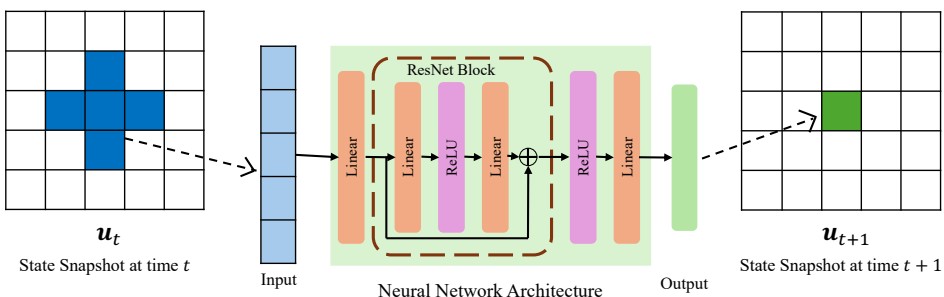

Figure 3: Residual network inspired neural network architecture used in our neural stencil emulator.

2. whether synthetic, PCA-guided designs improve few-shot generalization over on-trajectory and random space-filling baselines; and

3. whether hybrids that mix decorrelated on-trajectory stencils with synthetic/non-synthetic stencils offer the best robustness–efficiency trade-off; and

4. how sample-efficient NSE is relative to recent ML-based emulators, and how its long-horizon rollout accuracy and stability compares to these emulators?

NSE learns the discretized RHS from local inputs and is rolled out with an explicit integrator for forecasting. We evaluate on nonlinear PDEs–Allen-Cahn, Advection-Diffusion, and (scalar) Burgers'– to assess generalization to unseen initial conditions. Details of each PDE are in Appendix A.

**NSE architecture and training details.** Our NSE uses a two-block residual network (Fig. 3) with 64-unit linear layers. It predicts time derivative, advanced via explicit Euler timestepping. We train with Adam (initial learning rate 0.01) with cosine decay schedule for 5,000 epochs.

**Data and strategies.** For *pure* strategies, we use a short run of 10 timesteps from one initial condition on a $32 \times 32$ grid, yielding 10,240 stencils for *Short-Traj* strategy. *Random* strategies draw 10,240 stencils from the PDE's physically admissible range; *PCA-guided* strategies build 10,240 synthetic stencils by sampling in a principal-component space estimated from the short run and mapping them back to state space. For *mixed* strategies, we assume one full simulation of 1,000 timesteps over $t \in [0, 1]$ plus a small budget equivalent to 10 timesteps, which we allocate to (i) extending the trajectory (*Downsampled+Extend*), (ii) a short burst from a different initialization (*Downsampled+Diff Init*), or (iii) generating synthetic stencils (*Downsampled+Random*, *Downsampled+PCA*). In all mixed variants, we uniformly downsample the on-trajectory data to 10,240 stencils to reduce space–time correlation and combine them 1:1 with the supplemental batch of 10,240 synthetic/non-synthetic stencils yielding a total of 20,480 stencils. Under a small, fixed numerical solver budget, *mixed* strategies therefore train on twice as many stencils as *pure* strategies.

**Baselines.** We benchmark NSE against widely used ML-based emulator models: (1) U-Net neural-PDE surrogate (Takamoto et al., 2022), (2) Fourier Neural Operator (FNO) (Li et al., 2021), and (3) physics-informed neural network (PINN) (Raissi et al., 2019). Complete descriptions of the baseline implementations are provided in Appendix B.

**Evaluation protocol.** We take 10 unseen initial conditions and evolve each with NSE trained under the pure or mixed strategies above. Performance is reported as the trajectory of log–RMSE between 2D snapshots from the full-order numerical solver and NSE over the rollout horizon (Figures 4, 5, and 6). In Table 1, we report global NRMSE values for baselines along with their simulation data budget used in training. We downsample trajectories predicted by NSE in all sampling strategies by a factor of 10 for a fair comparison with baselines which are trained and tested on downsampled trajectories (refer to Appendix B). The expressions of the evaluation metrics are provided in Appendix C.

**Results.** Figures 4, 5, and 6 summarize Allen-Cahn, Advection-Diffusion, and Burgers', respectively, in a *2×3* layout: rows compare *Pure* vs. *Mixed* strategies; columns sweep diffusion coefficients $D \in \{5 \times 10^{-4}, 10^{-3}, 2 \times 10^{-3}\}$ from left to right. In Allen–Cahn system (Fig. 4), *PCA*-guided designs consistently outperform *Short-Traj* within the pure setting. Pure *Random* designs outperform both *PCA* and *Short-Traj* across all $D$ values–especially at longer horizons where coverage of rare stencil configurations matters; within the Random and PCA families, Sobol' low-discrepancy sampling typically slightly outperforms i.i.d. Uniform sampling.

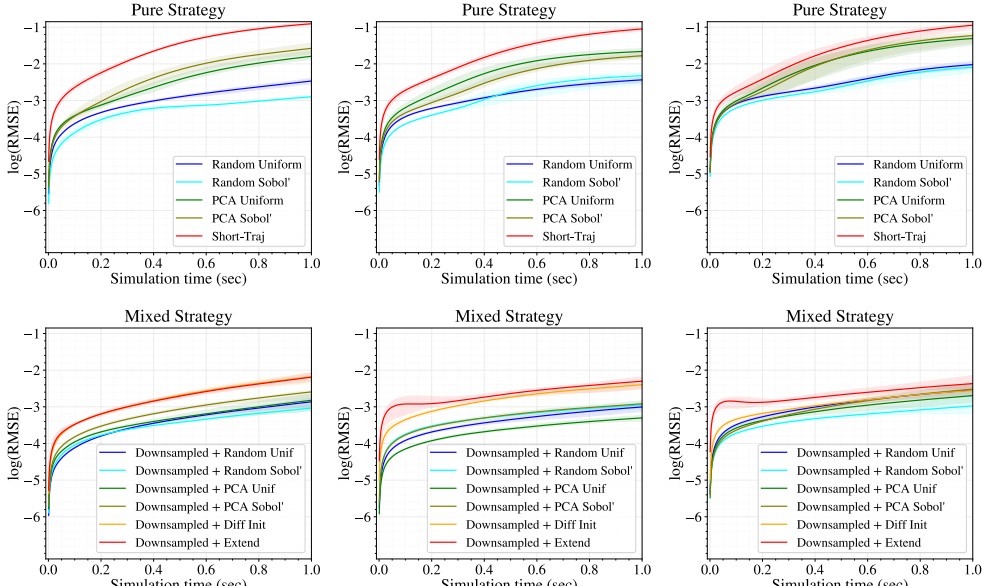

Figure 4: **Allen–Cahn system:** Neural stencil emulator's PDE rollout errors across different strategies using 3 diffusion coefficients. Columns (left→right) use $D \in \{5 \times 10^{-4}, 10^{-3}, 2 \times 10^{-3}\}$; rows compare *Pure* vs. *Mixed* sampling strategies. Curves aggregate NSE solutions over 10 unseen initial conditions providing mean of *log*-RMSE and the bands show 2-$\sigma$ variability around that mean.

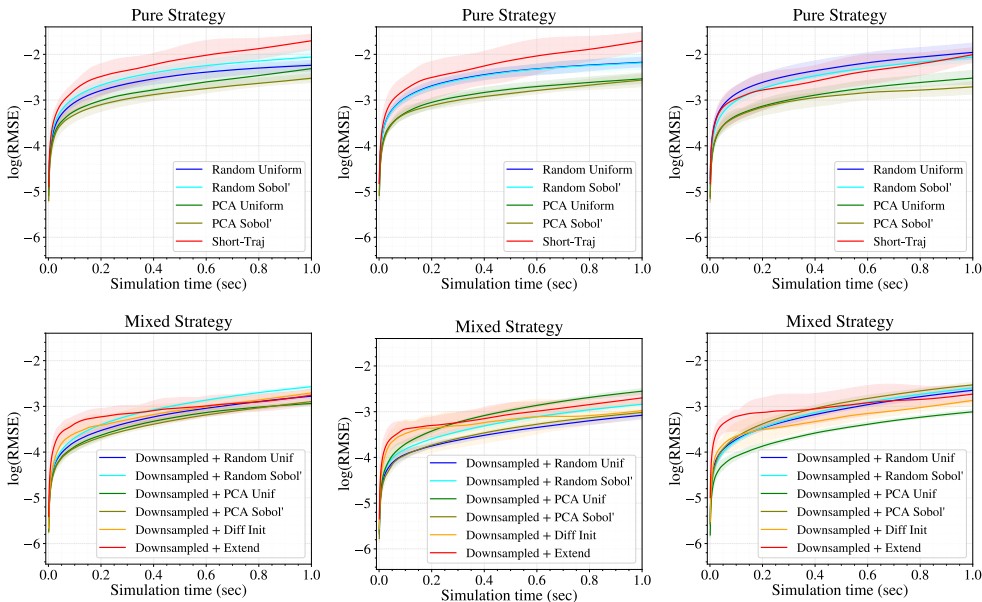

Figure 5: **Advection–Diffusion system:** Neural stencil emulator's PDE rollout errors across different strategies using 3 diffusion coefficients. Columns (left→right) use $D \in \{5 \times 10^{-4}, 10^{-3}, 2 \times 10^{-3}\}$; rows compare *Pure* vs. *Mixed* sampling strategies. Curves aggregate NSE solutions over 10 unseen initial conditions providing mean of *log*-RMSE and the bands show 2-$\sigma$ variability around that mean.

In the mixed setting, *Downsampled+PCA* shows clear, substantial gains over its pure *PCA* counterpart across all $D$ values. *Downsampled+Diff Init* and *Downsampled+Extend* also improve relative to their pure variants but generally trail *Downsampled+PCA*. *Downsampled+Random* offers only marginal gains over pure *Random*, likely because it does not exploit relationships in the on-trajectory data and downsampled stencils alone provide limited additional coverage. Overall, *Downsampled+PCA* and *Downsampled+Random* outperform *Downsampled+Diff Init* and *Downsampled+Extend*; within the mixed setting, we do not observe a consistent separation between Sobol' and Uniform variants.

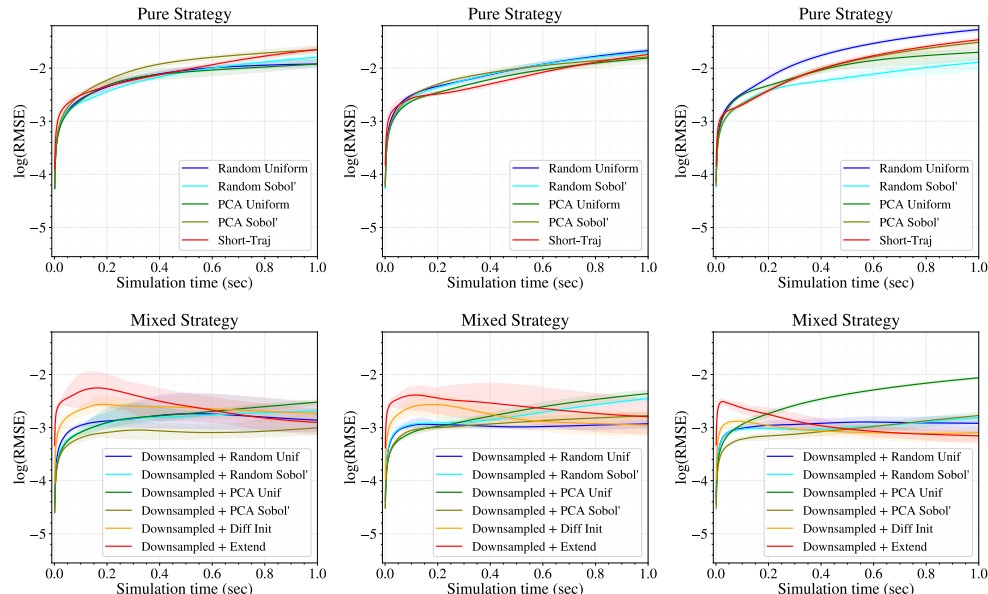

Figure 6: **Burgers' equation:** Neural stencil emulator's PDE rollout errors across different strategies using 3 viscosity coefficients. Columns (left→right) use $\nu \in \{5 \times 10^{-4}, 10^{-3}, 2 \times 10^{-3}\}$; rows compare *Pure* vs. *Mixed* sampling strategies. Curves aggregate NSE solutions over 10 unseen initial conditions providing mean of *log*-RMSE and the bands show 2-$\sigma$ variability around that mean.

*Why does Downsampled+PCA help?* In mixed strategies, PCA is fit to a full trajectory (not a short burst), capturing a broader basis of stencil states and their correlations; the synthesized stencils then better target rare but dynamically important directions, yielding consistent gains over pure *PCA* and other mixed baselines. Uniform downsampling (*Extend/Diff Init*) adds little span, so their improvements are modest over *Short-Traj*. In the pure setting, PCA's basis is narrow due to short run data–hence it performs worse than *Random* yet still better than *Short-Traj*.

In Advection-Diffusion system (Fig. 5), the *Pure* setting shows a clear ordering: *PCA*-guided designs are the best across the diffusion sweep, with *PCA-Sobol'* typically lowest error, followed closely by *PCA-Uniform*. Both *Random* variants trail the PCA designs, and *Short-Traj (Ergodic)* is consistently the worst. The separation is most pronounced at lower diffusion and narrows as $D$ increases (rightward columns), but the *PCA* strategy's lead persists throughout the horizon.

In the *Mixed* setting, results are more nuanced. *Downsampled+PCA* does not uniformly dominate: depending on $D$, it is often among the top performers but is comparable to *Downsampled+Random*, with overlapping error bands for much of the rollout. *Downsampled+Diff Init* and *Downsampled+Extend* again lag the remaining strategies highlighting importance of data augmentation. Overall, mixed strategies narrow the performance spread seen in the pure case; rankings vary with $D$ and horizon length, and no consistent Sobol' vs. Uniform winner among the compared strategies emerges.

In scalar Burgers' equation (Fig. 6), the *Pure* panels show little separation across diffusion coefficients: trajectories and bands largely overlap, especially in the first two columns. In the last column, *Random-Sobol'* and *PCA-Uniform* exhibit slightly lower late-time errors. Overall, under our short-run budget, sampling choice has only a modest effect; long-horizon error appears to be dominated by shock dynamics rather than the specific pure sampler.

In the *Mixed* panels, error bands are visibly larger and rankings become diffusion-dependent. *Downsampled+PCA* does not uniformly dominate; *Downsampled+Random*, *Downsampled+Extend*, and *Downsampled+Diff Init* are often comparable with overlapping error bands. The spread widens at lower diffusion (stronger advection/sharper features); among the non-synthetic mixed variants, *Downsampled+Extend* and *Downsampled+Diff Init* frequently underperform relative to *Short-Traj*. These patterns suggest that the performance on Burgers' is sensitive to augmentation: injecting synthetic or off-trajectory stencils can shift the data distribution near shocks and amplify compounding rollout error. Overall, no single method consistently wins across columns in mixed setting.

Table 1: Performance of NSE compared to baseline emulators under different data-sampling strategies. The top section reports baselines; the middle and bottom sections report NSE with pure and mixed strategies, respectively. Values are shown as $10^3 \times$ NRMSE (lower is better); divide by $10^3$ to recover raw NRMSE. The best NRMSEs among all models and among our pure strategies in each column are specified in **boldface** and via underline, respectively. Simulation budget represents overall solver timesteps needed to generate training data for each model. DS abbreviates downsampled.

| Model | Allen-Cahn | | | Advection-Diffusion | | | Burgers' | | | Simulation Budget |
|---|---|---|---|---|---|---|---|---|---|---|
| | $D{=}5e{-}4$ | $D{=}1e{-}3$ | $D{=}2e{-}3$ | $D{=}5e{-}4$ | $D{=}1e{-}3$ | $D{=}2e{-}3$ | $\nu{=}5e{-}4$ | $\nu{=}1e{-}3$ | $\nu{=}2e{-}3$ | |
| PINN | $995.4_{33.0}$ | $1007.2_{24.3}$ | $999.5_{24.2}$ | $887.1_{21.4}$ | $875.8_{37.2}$ | $869.3_{31.8}$ | $1121.8_{47.8}$ | $1135.4_{31.7}$ | $965.0_{48.6}$ | $0$ |
| U-Net | $532.0_{13.3}$ | $639.3_{34.4}$ | $414.0_{26.1}$ | $813.2_{34.7}$ | $768.3_{43.0}$ | $679.6_{58.1}$ | $2178.9_{222.6}$ | $2017.5_{234.9}$ | $2082.6_{328.9}$ | $1\times10^5$ |
| FNO | $390.5_{16.7}$ | $353.0_{20.2}$ | $198.3_{24.5}$ | $416.5_{47.9}$ | $320.6_{49.2}$ | $457.1_{74.1}$ | $555.1_{35.9}$ | $562.6_{41.3}$ | $432.9_{45.1}$ | $1\times10^5$ |
| U-Net | $401.8_{7.5}$ | $218.6_{9.1}$ | $166.4_{13.2}$ | $236.2_{19.7}$ | $174.7_{20.6}$ | $195.5_{25.3}$ | $902.4_{8.7}$ | $916.6_{16.8}$ | $945.0_{30.4}$ | $2\times10^6$ |
| FNO | $112.3_{12.1}$ | $106.7_{8.2}$ | $74.5_{10.8}$ | $120.4_{18.6}$ | $85.2_{11.3}$ | $54.6_{6.4}$ | $218.9_{13.1}$ | $229.8_{18.3}$ | $142.7_{12.1}$ | $2\times10^6$ |
| Random Uniform | $1.8_{0.2}$ | $\underline{2.3}_{0.2}$ | $5.5_{0.5}$ | $9.5_{1.5}$ | $12.1_{2.0}$ | $18.0_{4.0}$ | $133.0_{9.4}$ | $180.9_{20.0}$ | $438.5_{66.8}$ | $10$ |
| Random Sobol' | $\underline{0.8}_{0.1}$ | $2.7_{0.5}$ | $\underline{4.6}_{0.5}$ | $13.2_{2.3}$ | $11.7_{1.6}$ | $14.2_{2.3}$ | $145.5_{16.1}$ | $172.3_{13.8}$ | $\underline{114.6}_{13.0}$ | $10$ |
| PCA Uniform | $6.5_{1.0}$ | $12.4_{1.8}$ | $25.2_{4.7}$ | $6.1_{1.4}$ | $5.1_{0.8}$ | $5.2_{1.0}$ | $\underline{125.3}_{11.1}$ | $\underline{137.2}_{13.8}$ | $190.0_{21.7}$ | $10$ |
| PCA Sobol' | $11.0_{1.9}$ | $8.3_{1.0}$ | $28.1_{5.0}$ | $\underline{4.4}_{0.6}$ | $\underline{4.3}_{0.7}$ | $\underline{4.1}_{0.9}$ | $219.8_{12.9}$ | $155.9_{15.8}$ | $248.0_{42.3}$ | $10$ |
| Short-Traj | $56.8_{2.9}$ | $42.7_{3.8}$ | $53.1_{7.0}$ | $21.3_{8.0}$ | $20.7_{10.0}$ | $11.5_{4.6}$ | $183.7_{21.6}$ | $138.8_{22.5}$ | $266.6_{44.6}$ | $10$ |
| DS+Random Uniform | $0.7_{0.1}$ | $0.7_{0.1}$ | $2.0_{0.3}$ | $2.2_{0.1}$ | $\mathbf{1.2}_{0.1}$ | $3.2_{0.3}$ | $19.3_{10.2}$ | $\mathbf{15.3}_{1.3}$ | $16.6_{1.6}$ | $1.01\times10^3$ |
| DS+Random Sobol' | $\mathbf{0.5}_{0.1}$ | $0.8_{0.1}$ | $\mathbf{0.8}_{0.2}$ | $3.3_{0.3}$ | $2.0_{0.1}$ | $3.5_{0.4}$ | $23.8_{3.1}$ | $30.8_{3.8}$ | $16.3_{1.9}$ | $1.01\times10^3$ |
| DS+PCA Uniform | $0.7_{0.1}$ | $\mathbf{0.4}_{0.0}$ | $1.4_{0.2}$ | $1.7_{0.1}$ | $3.5_{0.4}$ | $\mathbf{1.2}_{0.1}$ | $29.5_{3.7}$ | $37.4_{5.0}$ | $76.0_{10.7}$ | $1.01\times10^3$ |
| DS+PCA Sobol' | $1.3_{0.1}$ | $0.9_{0.1}$ | $1.9_{0.3}$ | $\mathbf{1.6}_{0.1}$ | $1.4_{0.1}$ | $4.2_{0.5}$ | $\mathbf{11.5}_{2.0}$ | $19.1_{2.6}$ | $16.0_{2.4}$ | $1.01\times10^3$ |
| DS+Diff Init | $3.1_{0.4}$ | $2.6_{0.2}$ | $2.1_{0.3}$ | $2.4_{0.4}$ | $1.7_{1.0}$ | $2.0_{0.4}$ | $29.0_{8.5}$ | $20.0_{5.9}$ | $\mathbf{11.9}_{0.9}$ | $1.01\times10^3$ |
| DS+Extend | $3.0_{0.3}$ | $3.5_{0.5}$ | $3.6_{0.6}$ | $2.6_{0.8}$ | $2.8_{0.6}$ | $3.3_{1.4}$ | $31.0_{12.9}$ | $28.8_{14.9}$ | $14.3_{2.5}$ | $1.01\times10^3$ |

Table 1 summarizes NRMSE results for our NSE approach under *Pure* and *Mixed* strategies and ML-based emulators. Using a very small simulation budget to generate stencil training data, NSE consistently outperforms PINNs, U-Nets, and FNOs on Allen–Cahn, Advection–Diffusion, and Burgers' PDE systems. We attribute this gap to three key factors: (1) **local, low-dimensional inputs.** NSE operates on compact stencils (5-point neighborhood) rather than full fields, reducing input dimensionality by orders of magnitude and filtering out irrelevant global context. This alleviates the curse of dimensionality faced by full-field surrogates; (2) **high sample efficiency.** Each spatial grid location, along with its stencil, at a given timestep, yields an independent training sample. For instance, under a budget of 100 (or 2,000) simulations equaling a total of $10^5$ (or $2\times10^6$) solver timesteps, full-field surrogates effectively obtain either 500 (or $10^4$) training samples (additional details in Appendix B), as each rollout step consumes an entire spatial field. In contrast, NSE leverages every local stencil update as a training example, resulting in roughly $10^4$ effective samples from a budget of only 10 timesteps; (3) **robust temporal integration**. NSE learns discretized local time derivatives rather than approximating a global functional of the PDE residual, as in PINNs. Although PINNs in principle do not require explicit training samples, their optimization landscape is notoriously ill-conditioned, especially for high-dimensional PDEs, leading to slow convergence and large residual errors. In contrast, NSE learns a simple, low-dimensional mapping aligned with the underlying time-stepping scheme, which makes training significantly more stable and yields more accurate long-horizon rollouts. Together, these three factors enable NSE to achieve consistently superior performance over baselines, especially in data-limited regimes where efficient sample utilization and model complexity become critical. Lastly, Appendix E provides visualizations of the emulated dynamics, qualitatively comparing NSE against ML-based emulators.

## 5 CONCLUSION

In this work, we propose space-filling, spatial-correlation-preserving, data-augmentation strategies that generate localized stencil data and train NSE to learn PDE governing equations in a non-intrusive and sample-efficient ($\sim$10 timesteps' worth of simulation) manner. Our data-augmented stencils enable few-shot learning and accurate forecasts across Allen-Cahn, Advection–Diffusion, and Burgers' PDE systems from limited snapshots under tight compute budgets. Operationalized via NSE, our approach consistently outperforms on-trajectory sampling, underscoring the value of independent, space-filling stencil sampling for robust modeling of computer simulations. Finally, using only 10 solver steps' worth of stencil data, our approach outperforms ML emulators (FNO, U-Net, PINN) trained on thousands of trajectories in long-horizon accuracy and stability. We discuss limitations of our approach in Appendix D.

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

## A  PDE Systems

### A.1  Allen-Cahn System

The following system of equations describes the Allen–Cahn formulation:

$$\frac{\partial u(\boldsymbol{x},t)}{\partial t} = D\nabla^2 u(\boldsymbol{x},t) + 5(u(\boldsymbol{x},t) - u(\boldsymbol{x},t)^3) \quad \text{on } \Omega, \tag{3a}$$

$$u(\boldsymbol{x},t_0) \sim \text{GP}\big(m(\boldsymbol{x}), k(\boldsymbol{x},\boldsymbol{x})\big) \quad \text{on } \Omega, \tag{3b}$$

$$\nabla u(\boldsymbol{x},t) = 0 \quad \text{on } \partial\Omega. \tag{3c}$$

Here $D$ is the diffusion coefficient and $5(u(\boldsymbol{x},t) - u(\boldsymbol{x},t)^3)$ represents the source term. The GP represents the Gaussian process, used to generate the initial spatial field $u(\boldsymbol{x},t_0)$, and a zero-gradient boundary condition is imposed on the boundary $\partial\Omega$.

### A.2  Advection-Diffusion System

The following system of equations describes the Advection–Diffusion formulation on the domain $\Omega = [0,L]^n$:

$$\frac{\partial u(\boldsymbol{x},t)}{\partial t} = -a\nabla u(\boldsymbol{x},t) + D\nabla^2 u(\boldsymbol{x},t) \quad \text{on } \Omega, \tag{4a}$$

$$u(\boldsymbol{x},t_0) \sim \text{GP}\big(m(\boldsymbol{x}), k(\boldsymbol{x},\boldsymbol{x})\big) \quad \text{on } \Omega. \tag{4b}$$

$$u(\boldsymbol{x} + Le_i,t) = u(\boldsymbol{x},t) \quad \forall \boldsymbol{x} \in \partial\Omega, i = 1,2,\ldots,n \tag{4c}$$

Here $D$ is the diffusion coefficient, $a$ is the advection velocity, and $e_i$ denotes the unit vector along the $i^{th}$ coordinate axis. The GP represents the Gaussian process, used to generate the initial spatial field $u(\boldsymbol{x},t_0)$, and a periodic boundary condition is imposed on the boundary $\partial\Omega$.

### A.3  Scalar Burgers' System

The following system of equations describes the Scalar Burgers' formulation on the domain $\Omega = [0,L]^n$:

$$\frac{\partial u(\boldsymbol{x},t)}{\partial t} = -\nabla\Big(\frac{u^2(\boldsymbol{x},t)}{2}\Big) + \nu\nabla^2 u(\boldsymbol{x},t) \quad \text{on } \Omega, \tag{5a}$$

$$u(\boldsymbol{x},t_0) \sim \text{GP}\big(m(\boldsymbol{x}), k(\boldsymbol{x},\boldsymbol{x})\big) \quad \text{on } \Omega. \tag{5b}$$

$$u(\boldsymbol{x} + Le_i,t) = u(\boldsymbol{x},t) \quad \forall \boldsymbol{x} \in \partial\Omega, i = 1,2,\ldots,n \tag{5c}$$

Here $\nu$ is the viscosity, $u$ is the scalar velocity, and $e_i$ denotes the unit vector along the $i^{th}$ coordinate axis. The GP represents the Gaussian process, used to generate the initial spatial field $u(\boldsymbol{x},t_0)$, and a periodic boundary condition is imposed on the boundary $\partial\Omega$.

## B  Baseline Architectures and Training Protocols

We benchmark our Neural Stencil Emulator (NSE) against three representative ML-based emulator approaches, using their implementations provided in PDEBench (Takamoto et al., 2022).

**U-Net.** We employ a standard 2D U-Net as a data-driven PDE surrogate. It follows the classic encoder–decoder design with skip connections, enabling effective capture of multi-scale spatial features.

**Fourier Neural Operator (FNO).** We employ FNO for its efficiency and strong performance in operator learning. By parameterizing kernels in Fourier space and leveraging FFTs, FNO models long-range dependencies, supports mesh invariance, and generalizes across resolutions.

**Physics-Informed Neural Networks (PINNs).** We employ PINNs as physics-guided surrogates that embed PDE residuals directly into the loss function. This design enforces physical consistency while enabling data-efficient learning under sparse observations.

We employed the trajectories of length 1,000 timesteps for training and testing – consistent with our NSE setup – but downsampled them by a factor of 10 to sequences of 100 timesteps. This reduction was necessary because the baseline surrogates struggle to converge and remain stable over very long-horizons, where error accumulation prevents convergence. PINNs were trained directly on the full spatio–temporal mesh, enforcing the governing equations at all grid points. U-Net and FNO are trained autoregressively by minimizing prediction error over 20 rollout timesteps. We consider two simulation budgets: 100 or 2,000 numerical simulations, corresponding to $10^5$ and $2 \times 10^6$ solver timesteps, respectively. After downsampling each trajectory to 100 steps, we extract five non-overlapping 20-step subsequences $\{[1, 20], [21, 40], [41, 60], [61, 80], [81, 100]\}$, yielding 500 or $10^4$ training samples overall. For evaluation, we first downsample 10 held-out trajectories by a factor of 10 to generate trajectories of 100 timesteps and then roll out U-Net and FNO for the full 100 timesteps. PINNs are not rolled out autoregressively; instead, we assess them by comparing their predicted solutions over the mesh to ground-truth simulations.

## C    EVALUATION METRICS

To assess the accuracy and stability in long-horizon autoregressive rollout, we rely on (1) log root-mean-squared-error (log-RMSE) and (2) normalized RMSE (NRMSE). We compute RMSE for each state snapshot $\boldsymbol{u}_t$ at time $t$ on a 2D grid $N \times N$ as follows:

$$\mathrm{RMSE}_t = \frac{\|\hat{\boldsymbol{u}}_t - \boldsymbol{u}_t\|_F}{N},$$

where $\|.\|_F$ is the Frobenius norm. In Fig. 4, 5, and 6, we report $\log_{10}(\mathrm{RMSE}_t)$ over time. For numerical robustness, when plotting $\log_{10}(\mathrm{RMSE}_t)$, a small $\varepsilon$ (e.g., $10^{-12}$) may be added inside the logarithm. Next, we provide the expression for the scale invariant global NRMSE reported in Table 1. Specifically, NRMSE at time $t$ is calculated as:

$$\mathrm{NRMSE}_t = \frac{\|\hat{\boldsymbol{u}}_t - \boldsymbol{u}_t\|_F}{\|\boldsymbol{u}_t\|_F}.$$

From this, we compute the global NRMSE by uniformly averaging the normalized RMSEs over time.

## D    LIMITATIONS

We assume a model whose dynamics are fully governed by a stencil operator; training is performed offline; we do not enforce structure-preserving constraints (e.g., symmetries or conservation laws) on the learned stencil; and our data-augmentation strategies are static (i.e., no dynamic/active learning). Furthermore, NSE targets stencil-based computer model system identification, but standardized benchmarks with ground-truth operators and physics-aware diagnostics are scarce. Our primary evaluation–rollout error (characterized by log-RMSE, NRMSE, and absolute errors) on held-out initial conditions–provides useful signal yet only indirectly reflects physical fidelity (e.g., conservation, shock resolution, boundary fluxes). Although non-intrusive, NSE remains coupled to the discretization and time integrator; modeling and integration errors may be conflated, and transferability across grids, boundary conditions, and to 3D or multiphysics settings remains untested. Developing standardized operator-identification benchmarks and richer, physics-grounded metrics is an important future direction of work.

## E    VISUALIZATIONS OF EMULATED DYNAMICS

In Figures 7-12, we show ground-truth trajectories (from the test set of 10 trajectories) alongside the corresponding best (overall or out of *Pure* strategies) performing NSE model's autoregressive rollout predictions and their absolute errors, over 1,000 timesteps for all three PDE systems (for single diffusion or viscosity coefficients). In every case, NSE maintains high accuracy on an unseen initial condition over the entire rollout horizon.

**Allen–Cahn** system exhibits curvature-driven coarsening with interfaces that thicken and merge. NSE preserves the geometry and motion of these phase boundaries over entire rollout; the error remains small and concentrates along moving interfaces rather than diffusing across the domain. This

holds for both the overall best strategy, *Downsampled + PCA-Uniform* (Fig. 7), and the best *Pure* strategy, *Random-Uniform* (Fig. 8), closely matching solver snapshots throughout 0-1.0 seconds. The maximum absolute error of the overall-best NSE is $\sim 9\times$ lower than that of the best *Pure* strategy NSE, highlighting the usefulness of a single full-order simulation.

In **Advection–Diffusion** system, NSE tracks the phase-accurate transport of smooth modes while accurately capturing the gradual diffusion-driven amplitude decay. The absolute errors appear as thin, wave-like ridges aligned with advection structures and remain bounded over time in both the overall best strategy, *Downsampled + Random-Uniform* (Fig. 9), and the best *Pure* strategy, *PCA-Sobol'* (Fig. 10). Here, the maximum absolute error is $\sim 6.5\times$ lower for the overall-best NSE than for the best *Pure* strategy NSE, again underscoring the value of one full-order simulation.

**Burgers'** equation is the most visually demanding: nonlinear steepening generates sharp fronts and shock interactions. NSE maintains coherent shock locations as well as overall patterns over the full rollout horizon. As expected, errors are largest near emerging and interacting shocks, yet they do not trigger spurious oscillations or global drift. In comparison to the solver, discrepancies are confined to the steepest gradients, especially at later times, for both the overall best strategy, *Downsampled + Random-Uniform* (Fig. 11), and the best *Pure* strategy, *Random-Uniform* (Fig. 12). The maximum absolute error is $\sim 3\times$ lower for the overall-best NSE than for the best *Pure* strategy NSE, reinforcing the benefit of even a single full-order simulation.

Next, to complement the quantitative results provided in Table 1, we provide qualitative comparisons of predicted dynamics and corresponding absolute error maps for U-Net, FNO, and Best NSE (overall or out of *Pure* strategies). We omit PINN from these visualizations because its large error fields made the comparisons less informative and reduced the contrast needed to highlight differences among other models. Figures 13, 14, and 15 visualize the predicted trajectories alongside the ground truth for Allen–Cahn, Advection–Diffusion, and Burgers' PDE systems respectively. For each model, we additionally show the spatial absolute error fields. We take the square root for the error field, which does not change the relative distribution of errors but improves visual contrast, making differences between emulated dynamics by different models more distinguishable.

The visualizations reveal markedly different error patterns across different models. **First**, U-Net captures coarse spatial structures but fails to preserve amplitude fidelity, producing visually under-scaled rollouts. The error maps show structured regions of bias that accumulate rapidly with time, reflecting its inability to stabilize predictions over long-horizon rollouts. **Second**, FNO better maintains global amplitudes but exhibits frequency-dependent artifacts. Its error maps often display wave-like or grid-aligned patterns, particularly evident at intermediate and late timesteps. This suggests that certain spectral modes – especially high-frequency components – are poorly represented, consistent with the Fourier parametrization that favors low-frequency dynamics with high amplitude. **Third**, NSE demonstrates more balanced and stable behavior. NSE pure model (represents best *Pure* strategy) already reduces systematic biases and yields more homogeneous error maps. However, for challenging systems such as Burgers', faint oscillatory errors can still be observed, gradually accumulating with time and indicating residual long-horizon instability under limited sampling strategies. In contrast, NSE Overall (represents overall best sampling strategy), which effectively incorporates data selection, further suppresses these accumulative effects and produces nearly structureless error fields even at the end of rollouts. These findings suggest that our local downsampled stencils, when combined with auxiliary batch of stencils, not only reduce redundancy but also enhance stability and predictive accuracy, while – by virtue of their structural design – providing the flexibility needed to reliably identify and represent diverse dynamical regimes.

## F  LLM USAGE STATEMENT

The paper was edited for grammar and style using OpenAI's ChatGPT-5. The authors provided the drafted text and iteratively revised the AI-generated suggestions. All substantive content, methodology, experimental results, their discussion, and conclusions were developed solely by the human authors, who are fully responsible for the paper's final content.

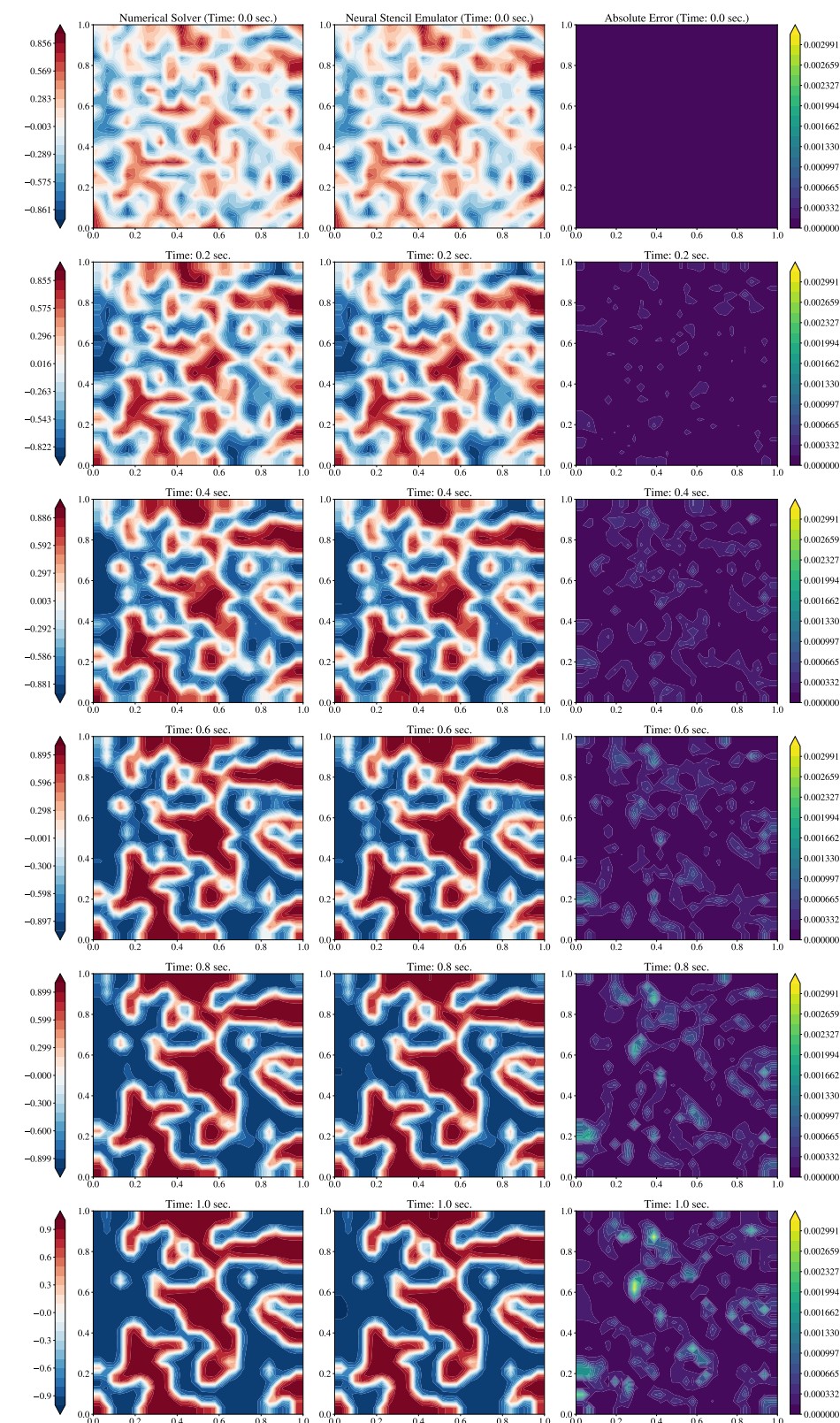

Figure 7: **Allen-Cahn system** ($D = 1 \times 10^{-3}$): the overall best sampling strategy in our NSE approach of *Downsampled + PCA-Uniform* strategy. NSE maintains highly accurate and stable predictions for rollout of unseen initial state over 1000 timesteps.

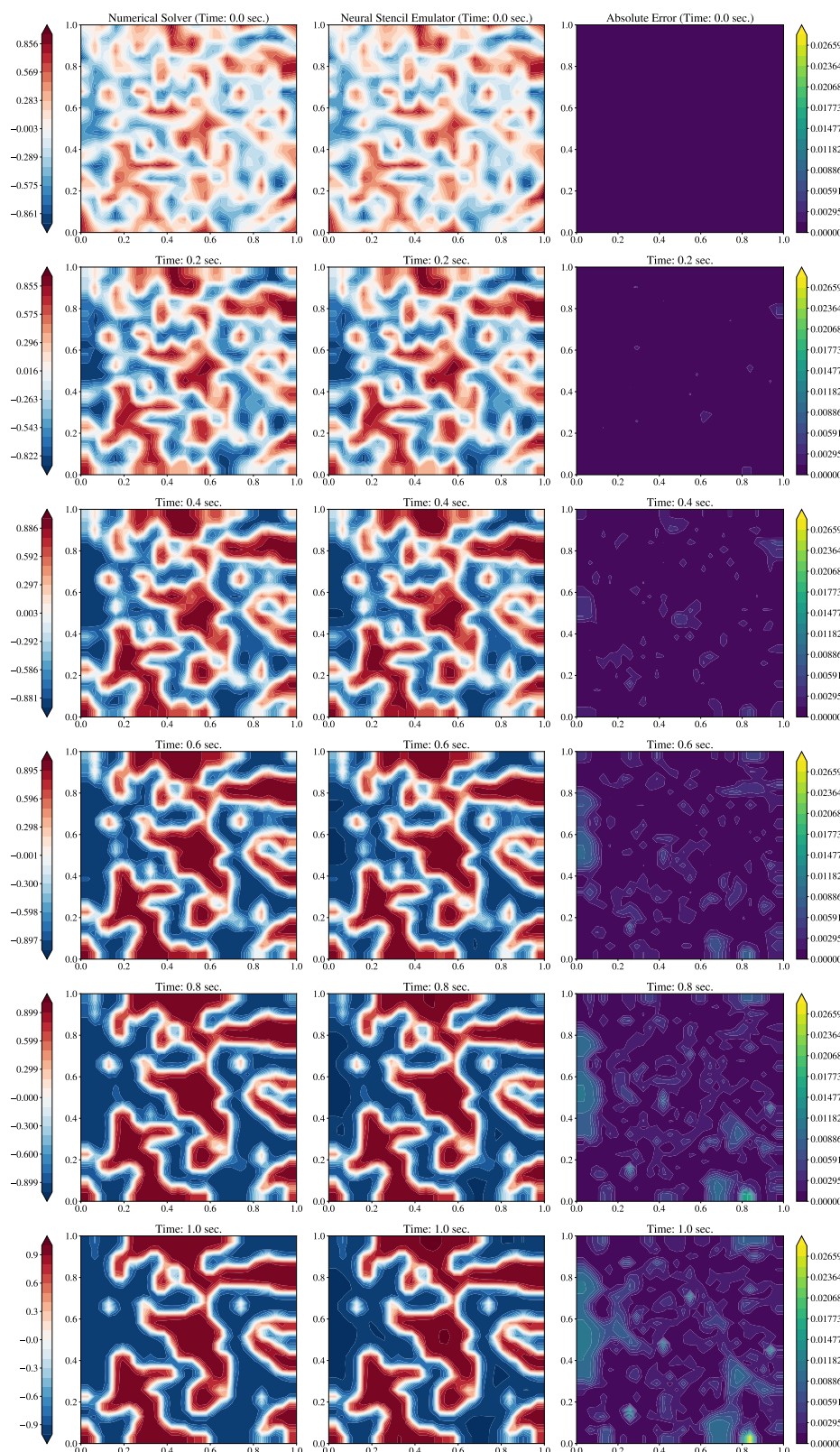

Figure 8: **Allen-Cahn system** ($D = 1 \times 10^{-3}$): the best *Pure* sampling strategy in our NSE approach of *Random-Uniform* strategy. Here, NSE trained on just 10 timesteps maintains highly accurate and stable predictions for rollout of unseen initial state over 1000 timesteps.

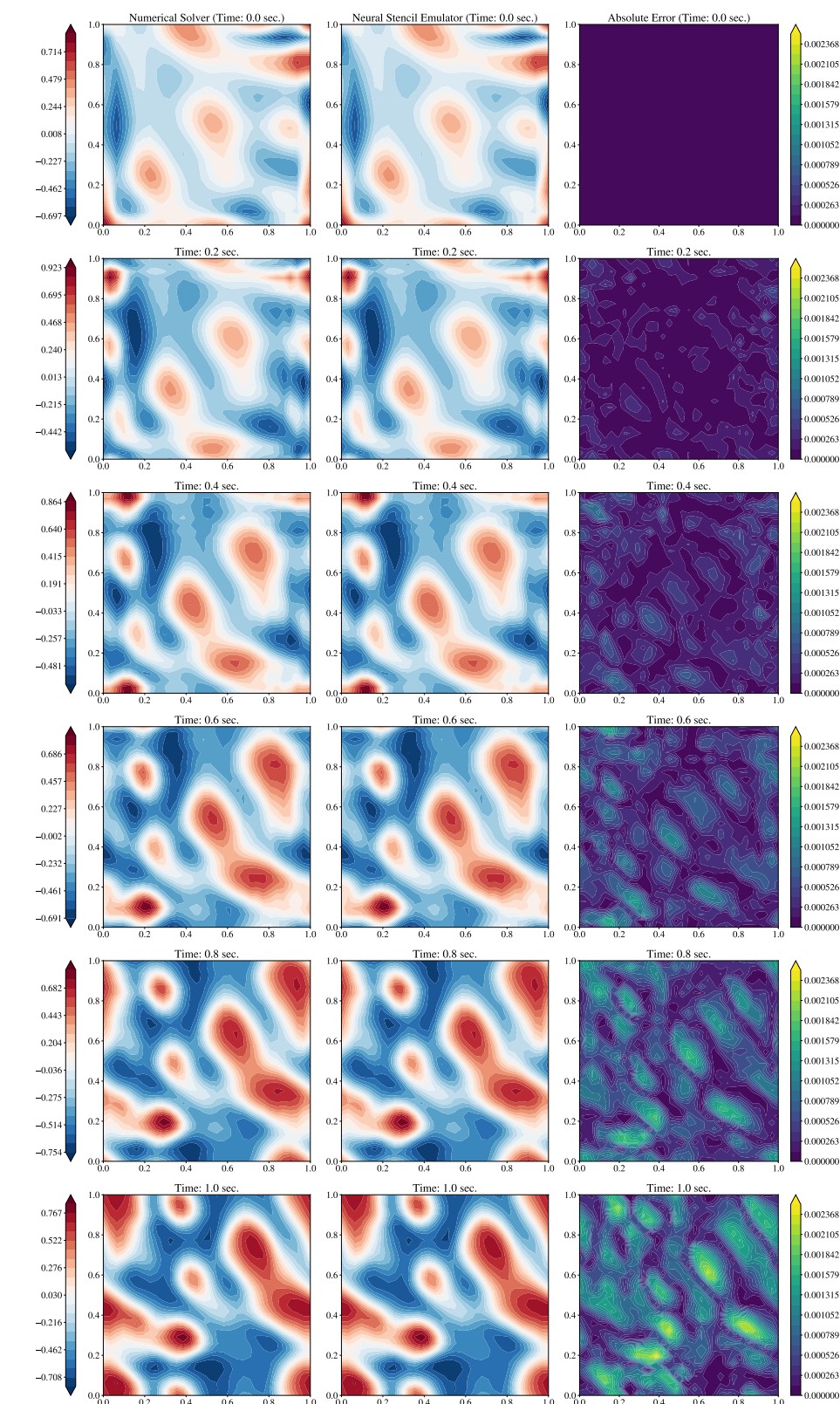

Figure 9: **Advection-Diffusion system** ($D = 1 \times 10^{-3}$): the overall best sampling strategy in our NSE approach of *Downsampled + Random-Uniform* strategy. NSE maintains highly accurate and stable predictions for rollout of unseen initial state over 1000 timesteps.

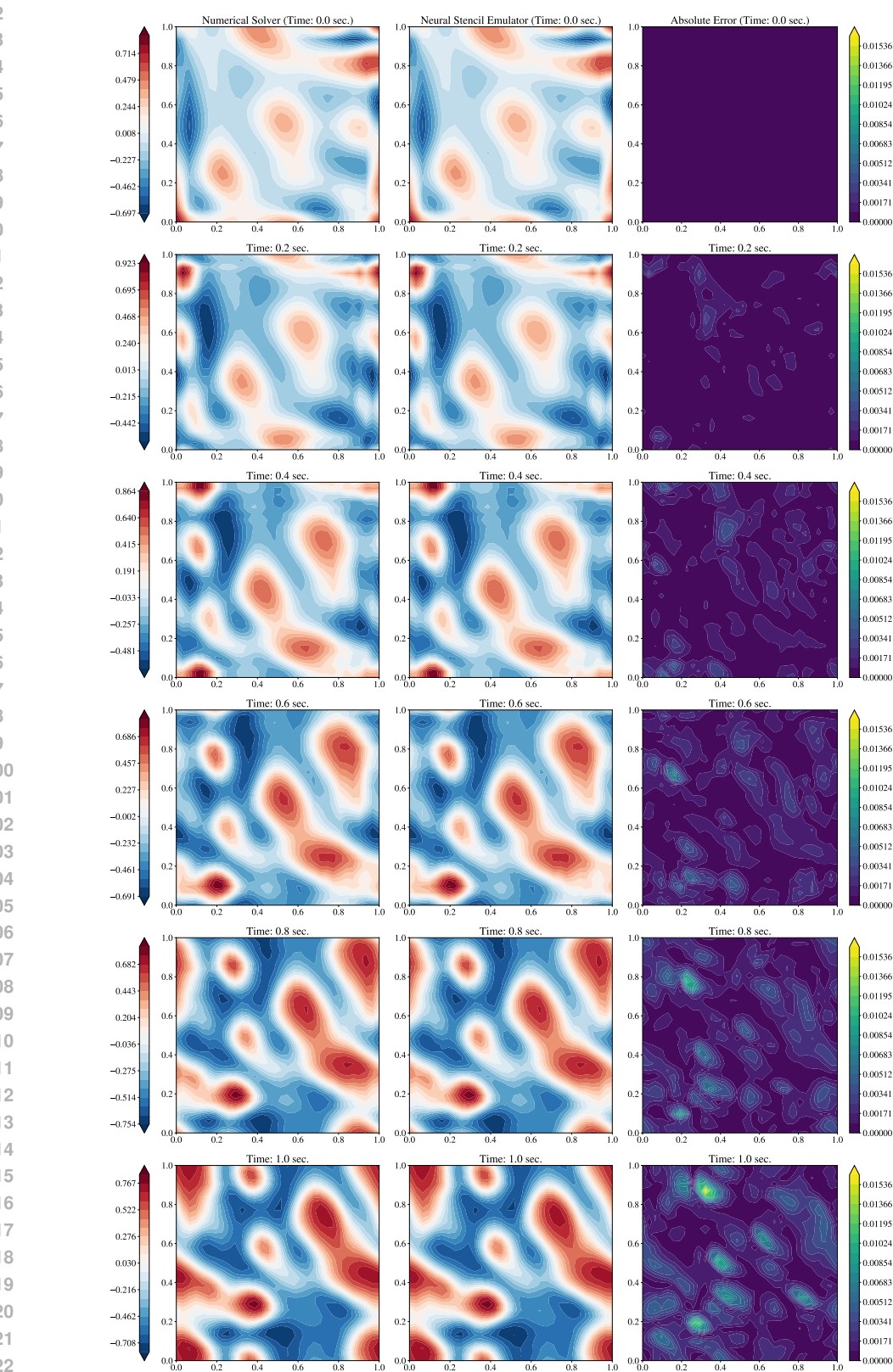

Figure 10: **Advection-Diffusion system** ($D = 1 \times 10^{-3}$): the best *Pure* sampling strategy in our NSE approach of *PCA-Sobol'* strategy. Here, NSE trained on just 10 timesteps maintains highly accurate and stable predictions for rollout of unseen initial state over 1000 timesteps.

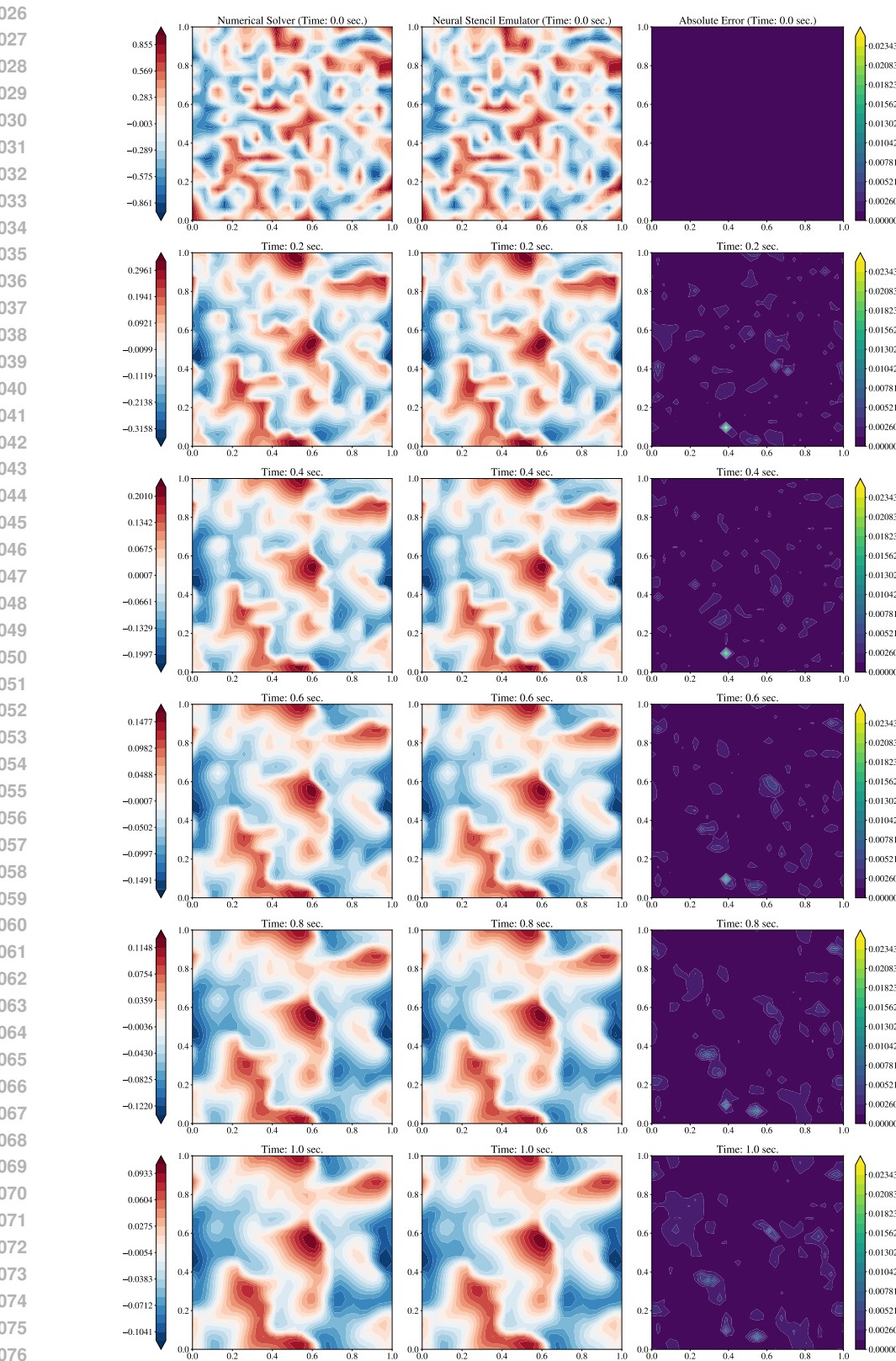

Figure 11: **Burgers' equation** ($\nu = 1 \times 10^{-3}$): the overall best sampling strategy in our NSE approach of *Downsampled + Random-Uniform* strategy. NSE maintains highly accurate and stable predictions for rollout of unseen initial state over 1000 timesteps.

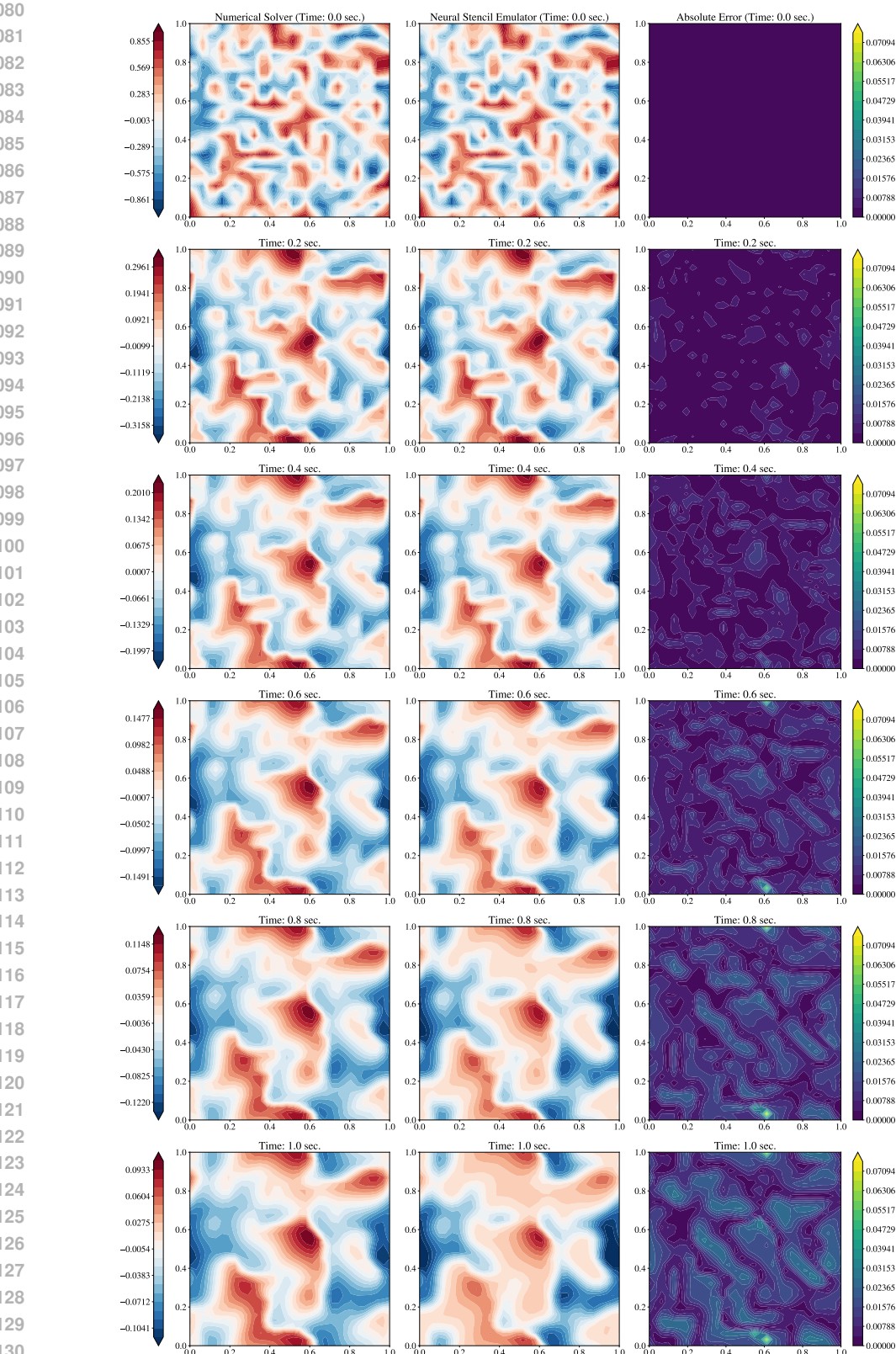

Figure 12: **Burgers' equation** ($\nu = 1 \times 10^{-3}$): the best *Pure* sampling strategy in our NSE approach of *Random-Sobol'* strategy. Here, NSE trained on just 10 timesteps maintains highly accurate and stable predictions for rollout of unseen initial state over 1000 timesteps.

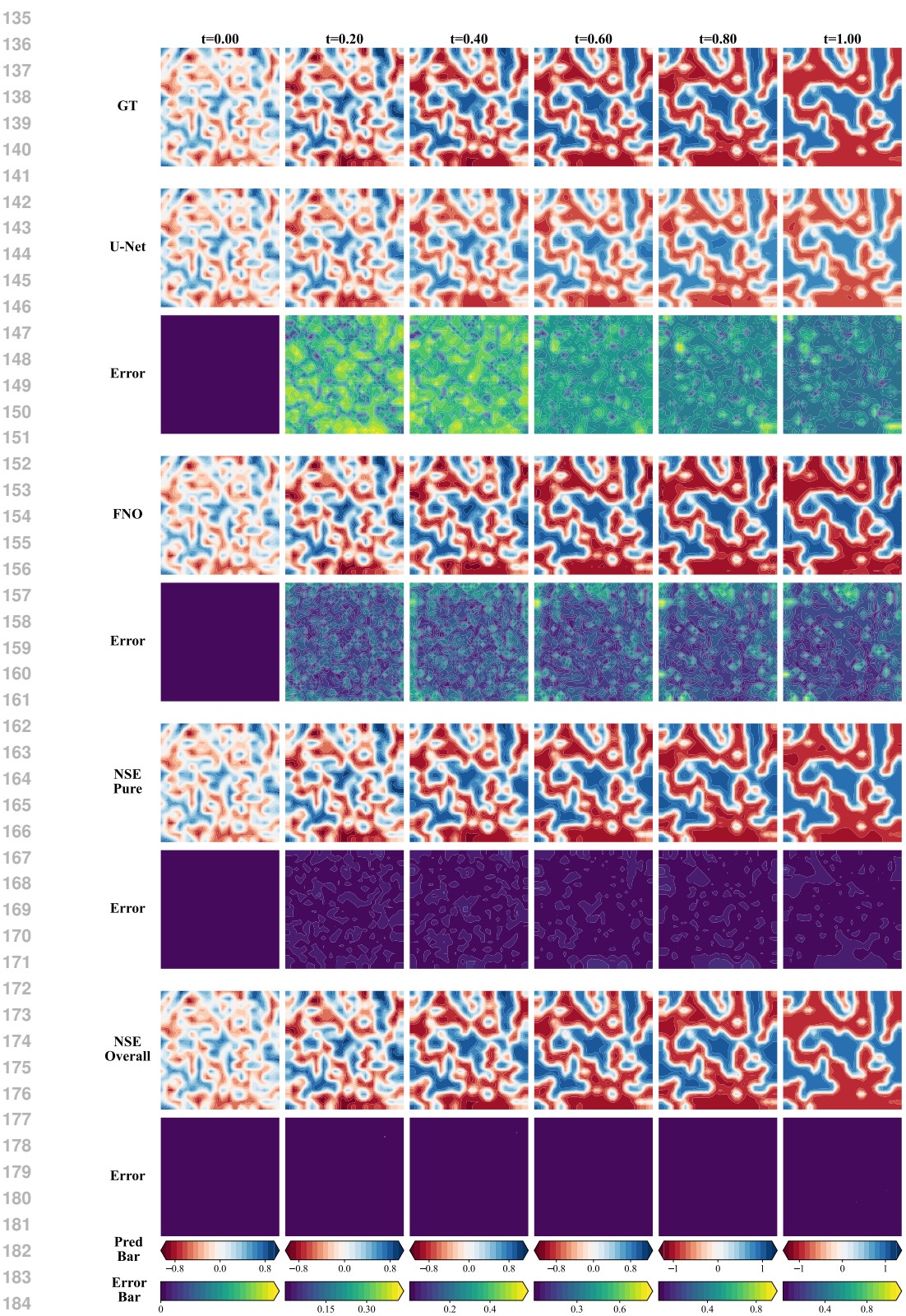

Figure 13: Visualizations of predicted dynamics and error maps for Allen–Cahn system ($D = 1 \times 10^{-3}$).

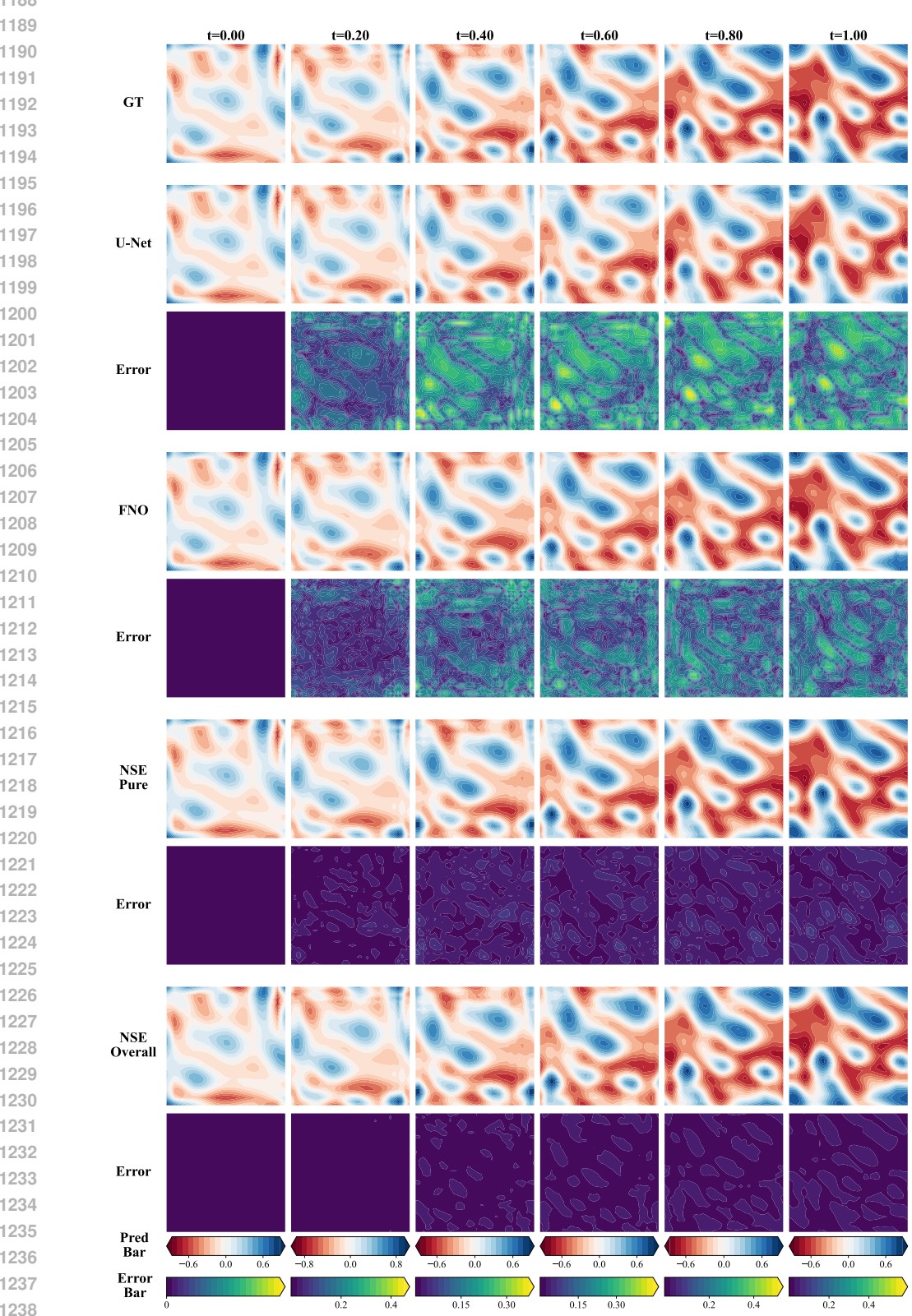

Figure 14: Visualizations of predicted dynamics and error maps for Advection–Diffusion system ($D = 1 \times 10^{-3}$).

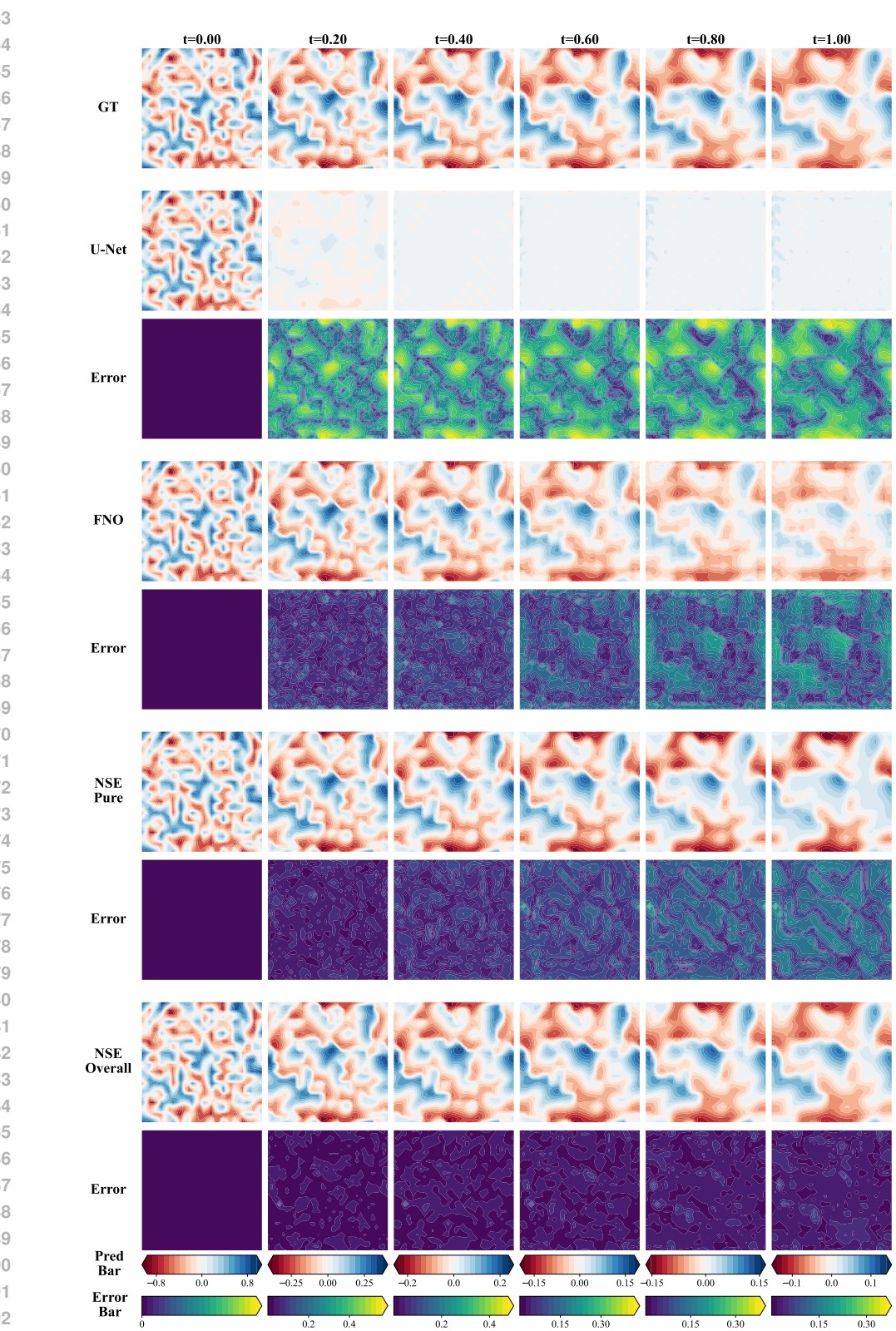

Figure 15: Visualizations of predicted dynamics and error maps for Burgers' equation ($\nu = 1 \times 10^{-3}$).

