# OpenReview forum: "Data-Augmented Few-Shot Neural Emulator for Computer-Model System Identification"
_ICLR.cc/2026/Conference — Submitted to ICLR 2026_

### Official Review · Reviewer_BdFj · 2025-10-27

**Soundness:** 1
**Presentation:** 2
**Contribution:** 1
**Rating:** 2
**Confidence:** 4

**Summary:**

The paper introduces a Neural Stencil Emulator and proposes a sample-efficient data augmentation strategy for generating PDE training data. The proposed scheme is evaluated on the Allen–Cahn, Advection–Diffusion, and Burgers’ equations.

**Strengths:**

The paper is well-motivated in its proposal of a sample-efficient training strategy for the model.
The authors also develop and explore a variety of data-augmentation and sampling schemes.

**Weaknesses:**

1. The paper mentions “decorrelating stencil samples,” and from the experimental results, it appears that reducing correlation improves performance. However, this strategy is not well explained. For example, does this finding apply only to the proposed scheme? Could other methods also be affected by data correlation, and what is the rationale behind why decorrelation benefits this particular approach?

2. The paper compares four schemes: Downsampled + (Diff Init, Extend, Random, and PCA). It is not entirely clear why the authors chose to include the Diff Init and Extend schemes, as their results do not seem to demonstrate meaningful insights. In contrast, the most effective method, Downsampled + PCA, lacks sufficient explanation regarding its implementation details, the rationale for its superior performance, and its potential limitations.

3. Overall, the experimental results are insufficient. The authors conducted only one experiment comparing all four data augmentation schemes—with and without downsampling—and a comparison with three other baseline methods. More comprehensive numerical experiments are needed to illustrate the method’s use cases, flexibility, and limitations.

4. In Table 1, the results for the baseline methods are significantly worse than those of the proposed scheme across all augmentation settings. There is no plausible explanation provided for this discrepancy, especially considering that the Advection–Diffusion example should not pose substantial difficulty for the baseline methods.

**Questions:**

Could the authors clarify whether the proposed NSE is an original method introduced in this paper, or if it was inspired by or derived from prior work? I did not find an explicit explanation of this in the paper. It would also be helpful if the authors could elaborate on how NSE differs from existing approaches that use stencils or local data to approximate PDE updates.

---

> ### Author Response · Authors · 2025-11-23
> **Response I**
>
> **Weakness 1** _The paper mentions “decorrelating stencil samples,” and from the experimental results, it appears that reducing correlation improves performance. However, this strategy is not well explained. For example, does this finding apply only to the proposed scheme? Could other methods also be affected by data correlation, and what is the rationale behind why decorrelation benefits this particular approach?_
>
> Our use of “decorrelating stencil samples” refers to breaking the strong space-time dependence you get from a single long simulation trajectory, where neighboring grid cell states are almost identical across space and time. From a statistical-learning standpoint, this is not specific to NSE: for any model, N more nearly independent samples carry more information than N highly correlated ones. What is special in our setting is that a stencil-based emulator permits us to actively sample arbitrary local states (via PCA or random designs), so we can approximate i.i.d. and space-filling coverage of the local state space in a way that pure trajectory-based methods cannot. Other surrogates could in principle also benefit from decorrelated data, but generating independent full-field samples is much more expensive than generating independent local stencils. We will clarify this intuition and the connection to our PCA / Random designs more explicitly in the final version.
>
> **Weakness 2** _The paper compares four schemes: Downsampled + (Diff Init, Extend, Random, and PCA). It is not entirely clear why the authors chose to include the Diff Init and Extend schemes, as their results do not seem to demonstrate meaningful insights. In contrast, the most effective method, Downsampled + PCA, lacks sufficient explanation regarding its implementation details, the rationale for its superior performance, and its potential limitations._
>
> We included the Downsampled+Extend and Downsampled+Diff-Init mixed strategies as diagnostic controls that expend the same small additional simulation budget in two natural yet simple ways–either by continuing the long reference trajectory or by starting a short new one from different initial condition, respectively–so we can test whether simply collecting more on-trajectory data and combining with downsampled stencils from existing reference trajectory could lead to any improvements. As our results show, both variants yield only similar improvements and generally lag behind the Random/PCA mixes, which highlights that the main benefit comes from decorrelated, off-trajectory stencil augmentation rather than from extra steps generated by numerical solver. By contrast, Downsampled+PCA first fits PCA to the full reference trajectory, constructs a bounding box in principal-component space, and samples synthetic stencils inside that box (with the help of rejection sampling which happens before evaluating such synthetic stencils through the numerical solver), so the extra budget is focused along dynamically important directions while staying within the empirically observed state manifold. Our additional simulation-budget ablation (varying the number of timesteps while always downsampling back to a fixed stencil count; also in our response to reviewer N5RJ’s question 3) further supports this interpretation: PCA-based designs are robust once they have seen roughly 100–200 steps and remain competitive regardless of reference trajectory length used to construct PCA-based stencils, whereas Short-Traj (but trained using same 10 timesteps worth of downsampled stencils) benefits almost monotonically as we use larger reference trajectory to downsample from. So, when we combine such downsampled stencils with synthetic PCA stencils we see further gains as shown in our paper. Whereas combining same downsampled stencils with on trajectory stencils in either Diff-Inti or Extend mixed strategies yields modest improvements. We will expand the description of Downsampled+PCA and its rationale in the final version for clarity.

---

> ### Author Response · Authors · 2025-11-23
> **Response II**
>
> **Weakness 3** _Overall, the experimental results are insufficient. The authors conducted only one experiment comparing all four data augmentation schemes—with and without downsampling—and a comparison with three other baseline methods. More comprehensive numerical experiments are needed to illustrate the method’s use cases, flexibility, and limitations._
>
> We respectfully disagree that the experimental evidence is insufficient. Our main results already span three distinct PDE families (Allen-Cahn, advection-diffusion, Burgers) with multiple diffusivity/viscosity settings, pure vs. mixed sampling strategies, and comparisons against three surrogate paradigms (PINNs, U-Net, FNO) under substantially larger (2 different) simulation budgets, all evaluated on 1,000-step rollouts over 10 unseen initial conditions. In the rebuttal phase we further added: (i) a mass-conservation ablation for 2D diffusion, (ii) a detailed simulation-budget sweep where we vary the number of solver steps but fix the stencil budget, and (iii) derivative-prediction FNO/U-Net baselines with explicit time integration, all of which consistently support our main claim that localized stencil-space augmentation enables strong few-shot surrogates. We believe the current suite of experiments already demonstrates both the usefulness and robustness of the proposed NSE in the few-shot regime.
>
> **Weakness 4** _In Table 1, the results for the baseline methods are significantly worse than those of the proposed scheme across all augmentation settings. There is no plausible explanation provided for this discrepancy, especially considering that the Advection–Diffusion example should not pose substantial difficulty for the baseline methods._
>
> You are right that the advection-diffusion case is not intrinsically “hard” for modern surrogates; the key discrepancy here comes from how data and parameterization are used, not from PDE difficulty. In Table 1, the baselines are actually given much larger simulation budgets to generate their training data than NSE (100 or 2,000 full 1,000-step trajectories, i.e., $10^5$ or $2\times10^6$ solver timesteps), whereas NSE’s pure strategies use only 10 timesteps (yielding 10,240 stencils) and the mixed strategies add just one extra long reference trajectory in addition to 10 timesteps’ worth of stencils.
>
> The baselines see these data as high-dimensional global snapshots and after downsampling and slicing into 20-step sub-sequences, effectively train on only hundreds to $10^4$ correlated samples; in contrast, NSE treats each local stencil update as an independent low-dimensional training example, so the same or smaller simulation budget yields orders of magnitude more statistically useful samples for training. Combined with the fact that FNO/U-Net must learn a global flow map that is then rolled out autoregressively (and thus accumulates error), while NSE learns a local discretized RHS aligned with the time-stepping scheme, this leads to substantially better long-horizon accuracy even on relatively simple systems like advection-diffusion.
>
> **Question 1** _Could the authors clarify whether the proposed NSE is an original method introduced in this paper, or if it was inspired by or derived from prior work? I did not find an explicit explanation of this in the paper. It would also be helpful if the authors could elaborate on how NSE differs from existing approaches that use stencils or local data to approximate PDE updates._
>
> Neural stencil emulators themselves are not new, and we do not claim otherwise–our paper cites prior work (Maddu et al. 2023) that uses neural networks to approximate local stencil updates. What we introduce here is NSE as a data-augmented, few-shot neural stencil emulator, specifically designed to work in the regime where few ( even as little as $\sim$10) timesteps from a single high-fidelity trajectory are available. Our main contribution is the localized stencil-space augmentation strategy (on-trajectory downsampling combined with PCA/random sampling) that turns one or a few trajectories into tens of thousands of diverse, approximately independent stencil examples. This enables stable, accurate surrogates under extremely limited simulation budgets where existing surrogate methods typically require many trajectories and initial conditions. We will clarify this positioning in the revised manuscript and more explicitly distinguish NSE from prior neural-stencil approaches.
>
> Maddu, S., et al. "STENCIL-NET for equation-free forecasting from data." Scientific Reports, 13(1), 2023.

---

### Official Review · Reviewer_Sfiu · 2025-10-28

**Soundness:** 2
**Presentation:** 2
**Contribution:** 1
**Rating:** 2
**Confidence:** 4

**Summary:**

The authors develop a data sampling/augmentation strategy and model to learn finite difference stencils. The method seems to work well for simple 2D PDEs, and the data augmentation aids in learning good stencils.

**Strengths:**

- In general, the paper is well written and the augmentation strategies are discussed well.
- Learning a local neural network approximation is interesting and not conventionally done.
- The work addresses a problem that a lot of PDE data is redundant, which is a real problem.

**Weaknesses:**

### Major Concerns

- The discussed baselines (FNO/Unet) should be reported using the same explicit time integrator as well. From prior work, it is known that for simple, 2D PDEs that are discretized finely in time (in your case you use 1000 timesteps), training networks to predict the current time derivative and using a temporal integrator is more effective (https://www.sciencedirect.com/science/article/pii/S0045782525002622). Reporting both the base Unet/FNO and the derivative Unet/FNO would give a better idea of where the improvement is coming from.
    - Related to this, the accuracy of models that are based on explicit time integrators is dependent on the timestep size chosen for inference. For more complex systems, this is not even stable (i.e., even with perfect estimates of the RHS, the explicit integrator will still accumulate error), and in general, brings back much of the discretization requirements that we are hoping to avoid by training a neural PDE surrogate.

- The tested cases are, in general, quite restrictive. The Allen-Cahn and Advection-Diffusion equations are relatively smooth, do not have shocks, and evolve slowly. This allows finite-difference based methods to work well. When transitioning to the Burgers equation, the presence of some shocks (although dampened by the viscosity) already makes the model perform an order of magnitude more poorly. When moving away from these toy problems into more complex phenomena (turbulence, instability, etc. (https://arxiv.org/abs/2412.00568)) I am not confident in this method.
    - Related to this point is that for more complex systems, having a 5-dimensional representation to make a prediction is likely no longer sufficient. Either a much higher resolution would be needed (more than 32x32) to resolve the relevant scales (such as in DNS) or a full-field method can use a current, coarsened field to approximate a future, coarsened field (although this is a rather difficult problem in general)
    - It also seems that the current method only works with periodic BCs (perhaps there is a way to approximate a one-sided stencil) and for regular grid problems (irregular meshes do not admit a consistent stencil)

### Minor Concerns
- There is marginally more cost by needing to evaluate a stencil pointwise across a field rather than predicting the full field at once with a Unet/FNO
- There are prior works (https://www.nature.com/articles/s41598-023-39418-6, https://arxiv.org/abs/2201.01854) that seek to approximate a local stencil with a neural network, which depending on your perspective, reduces the main contribution of the work to be a data augmentation strategy.

**Questions:**

- The dx and dt spacing should be made clear for the generated data and the training data. The paper says that the simulations are solved with a 32x32 grid, but that seems much too coarse for a numerical solver? I’m under the impression that the numerical simulator is run at a higher resolution and then downsampled spatially and (perhaps) temporally.

---

> ### Author Response · Authors · 2025-11-23
> **Response I**
>
> **Weakness 1** _The discussed baselines (FNO/Unet) should be reported using the same explicit time integrator as well. From prior work, it is known that for simple, 2D PDEs that are discretized finely in time (in your case you use 1000 timesteps), training networks to predict the current time derivative and using a temporal integrator is more effective. Reporting both the base Unet/FNO and the derivative Unet/FNO would give a better idea of where the improvement is coming from._
>
> We followed your suggestion and implemented derivative-prediction baselines for both U-Net and FNO, for **both** simulation budgets used in the main paper ($10^5$ and $2\times10^6$ solver timesteps). Concretely, we use the canonical architectures and train them either as standard one-step state predictors (“normal”) or to predict the current time derivative combined with an explicit Forward Euler integrator (“FwdEuler”), keeping the same 1,000-step rollout horizon and grid as in our NSE experiments. The tables below report $10^3\times$nRMSE (lower is better). For U-Net, derivative training with Euler generally reduces error relative to the corresponding state-prediction model at the smaller budget (with a couple of exceptions for Allen-Cahn), and at the larger budget it improves over “normal’’ U-Net on Allen-Cahn and advection-diffusion but degrades performance on Burgers; in all cases it still falls short of NSE, which uses a vastly smaller simulation budget. For FNO, derivative training yields only modest, problem-dependent changes–slight improvements on some diffusion settings but worse performance on Burgers at the smaller budget, and consistent underperformance relative to “normal’’ FNO across all three systems at the larger budget–again remaining above NSE errors. Overall, these results confirm that while derivative prediction plus an explicit integrator can sometimes help baseline models, it does not close the gap to our stencil-based approach, whose main advantage comes from exploiting locality and synthetic stencil augmentation in the few-shot regime.
>
> | Model  |  | Allen-Cahn |  |  | Adv-Diff |  |  | Burgers |  | Budget |
> | :-: | :-: | :-: | :-: | :-: | :-: | :-: | :-: | :-: | :-: | :-: |
> |  | D=5e-4 | D=1e-3 | D=2e-3 | D=5e-4 | D=1e-3 | D=2e-3 | ν=5e-4 | ν=1e-3 | ν=2e-3 |  |
> | U-Net (original) | 532.0$_{13.3}$ | 639.3$_{3.4}$ | 414.0$_{2.4}$ | 813.2$_{24.7}$ | 679.6$_{58.1}$ | 518.2$_{35.7}$ | 2178.9$_{22.7}$ | 2017.5$_{23.4}$ | 2082.6$_{29.2}$ | $1\times10^5$ |
> | **U-Net (normal)** | **774.0**$_{15.2}$ | **896.9**$_{27.6}$ | **718.5**$_{18.1}$ | **1109**$_{33.4}$ | **1191.6**$_{41.0}$ | **920.8**$_{29.7}$ | **1323**$_{35.5}$ | **975.6**$_{24.9}$ | **952.7**$_{27.3}$ | **$1\times10^5$** |
> | **U-Net (FwdEuler)** | **790.6**$_{14.3}$ | **736.7**$_{11.7}$ | **734.5**$_{13.5}$ | **1001**$_{28.9}$ | **804.5**$_{22.4}$ | **817.8**$_{19.1}$ | **932.4**$_{21.6}$ | **863.4**$_{18.7}$ | **900.6**$_{20.8}$ | **$1\times10^5$** |
> | FNO (original) | 390.3$_{16.7}$ | 353.0$_{20.2}$ | 198.3$_{4.5}$ | 416.5$_{49.2}$ | 260.6$_{49.2}$ | 457.1$_{74.1}$ | 555.1$_{35.9}$ | 562.6$_{41.3}$ | 432.9$_{45.1}$ | $1\times10^5$ |
> | **FNO (normal)** | **270.5**$_{9.8}$ | **253.0**$_{11.1}$ | **177.6**$_{8.6}$ | **377.5**$_{17.4}$ | **289.4**$_{14.2}$ | **422.4**$_{19.6}$ | **440.1**$_{16.0}$ | **372.1**$_{13.7}$ | **360.6**$_{14.9}$ | **$1\times10^5$** |
> | **FNO (FwdEuler)** | **264.0**$_{12.0}$ | **242.6**$_{10.7}$ | **192.5**$_{11.6}$ | **337.4**$_{15.2}$ | **287.8**$_{13.9}$ | **368.7**$_{17.0}$ | **536.3**$_{22.7}$ | **634.9**$_{26.4}$ | **480.3**$_{20.1}$ | **$1\times10^5$** |
> | ----- | ----- | ----- | ----- | ----- | ----- | ----- | ----- | ----- | ----- | ----- | ----- | ----- | ----- |
> | U-Net (original) | 401.8$_{7.5}$ | 218.6$_{9.1}$ | 166.4$_{13.2}$ | 236.2$_{19.7}$ | 174.7$_{20.6}$ | 195.5$_{25.3}$ | 902.4$_{8.7}$ | 916.6$_{16.8}$ | 945.0$_{30.4}$ | $2\times10^6$ |
> | **U-Net (normal)** | **439.3**$_{14.8}$ | **245.8**$_{9.7}$ | **197.5**$_{8.9}$ | **223.4**$_{11.2}$ | **263.9**$_{12.6}$ | **432.5**$_{18.4}$ | **609.3**$_{20.1}$ | **625.1**$_{21.7}$ | **613.1**$_{19.5}$ | **$2\times10^6$** |
> | **U-Net (FwdEuler)** | **285.2**$_{10.3}$ | **242.5**$_{8.4}$ | **168.6**$_{7.9}$ | **224.3**$_{10.8}$ | **228.6**$_{11.1}$ | **225.8**$_{9.6}$ | **622.9**$_{22.5}$ | **640.4**$_{23.8}$ | **740.4**$_{25.1}$ | **$2\times10^6$** |
> | FNO (original) | 112.3$_{12.1}$ | 106.7$_{8.2}$ | 74.5$_{10.8}$ | 120.41$_{8.6}$ | 85.2$_{11.3}$ | 54.6$_{6.4}$ | 218.9$_{13.1}$ | 229.8$_{13.3}$ | 142.7$_{12.1}$ | $2\times10^6$ |
> | **FNO (normal)** | **216.8**$_{7.9}$ | **205.4**$_{7.2}$ | **164.2**$_{6.5}$ | **132.9**$_{9.0}$ | **103.4**$_{6.1}$ | **79.6**$_{5.7}$ | **192.2**$_{10.1}$ | **226.8**$_{11.3}$ | **154.9**$_{9.4}$ | **$2\times10^6$** |
> | **FNO (FwdEuler)** | **224.0**$_{9.8}$ | **207.9**$_{8.4}$ | **183.6**$_{8.1}$ | **133.0**$_{11.2}$ | **111.6**$_{7.5}$ | **82.2**$_{6.8}$ | **247.7**$_{13.6}$ | **235.0**$_{14.4}$ | **156.7**$_{10.3}$ | **$2\times10^6$** |

---

> ### Author Response · Authors · 2025-11-23
> **Response II**
>
> **Weakness 1.1** _Related to this, the accuracy of models that are based on explicit time integrators is dependent on the timestep size chosen for inference. For more complex systems, this is not even stable (i.e., even with perfect estimates of the RHS, the explicit integrator will still accumulate error), and in general, brings back much of the discretization requirements that we are hoping to avoid by training a neural PDE surrogate._
>
> We agree that explicit integrator-based surrogates are sensitive to the timestep used at inference and can become unstable for very complex or stiff systems–even with accurate RHS estimates. However, this issue is not unique to derivative, timestepper-based surrogates: autoregressive full-field models can also exhibit error growth and instability over long horizons, and the broader neural PDE literature is actively exploring ways to mitigate this. In our experiments we operate in a finely discretized regime (1,000 small timesteps) and keep the timestep fixed and identical across all explicit baselines, so the comparison is fair. The focus of this paper is therefore not to remove all discretization-related constraints for every possible PDE, but to show that exploiting locality and homogeneity at the stencil level enables strong few-shot performance under tight simulation budgets.
>
> **Weakness 2** _The tested cases are, in general, quite restrictive. The Allen-Cahn and Advection-Diffusion equations are relatively smooth, do not have shocks, and evolve slowly. This allows finite-difference based methods to work well. When transitioning to the Burgers equation, the presence of some shocks (although dampened by the viscosity) already makes the model perform an order of magnitude more poorly. When moving away from these toy problems into more complex phenomena (turbulence, instability, etc.) I am not confident in this method._
>
> We agree that our current testbed (Allen-Cahn, advection-diffusion, viscous Burgers) does not cover the full spectrum of challenging PDEs such as fully turbulent or strongly unstable flows, and we are careful not to claim universal robustness in those regimes. However, our approach is not tied to finite-difference schemes or “toy” equations: the same stencil-emulation idea can be applied to finite-element and other spatial discretizations, as well as implicit or multi-stage time integrators. The common ingredient is exploiting locality and parameter sharing by learning a low-dimensional local operator whose input size is independent of the global domain or resolution. The goal of this paper is therefore not to propose a single surrogate that dominates in all complex regimes, but to demonstrate that leveraging locality via data-augmented stencil training can dramatically reduce simulation budget in few-shot training settings where only a small number of full trajectories are available, and other methods that perform well in a rich-data regime may degrade.
>
> **Weakness 2.1** _Related to this point is that for more complex systems, having a 5-dimensional representation to make a prediction is likely no longer sufficient. Either a much higher resolution would be needed (more than 32x32) to resolve the relevant scales (such as in DNS) or a full-field method can use a current, coarsened field to approximate a future, coarsened field (although this is a rather difficult problem in general)_
>
> NSE can accommodate larger spatial stencils (e.g., 13-point), to multi-time stencils (stacking several past steps), and to additional local features (e.g., coefficients, forcings). As long as the stencil size remains much smaller than the global domain--which is almost always true for practically useful simulation settings--the key advantages of our approach should persist: the input dimensionality of the learned operator scales only with stencil size (which is a fixed feature of the discretized dynamical system itself), not with grid resolution or domain extent.
>
> **Weakness 2.2** _It also seems that the current method only works with periodic BCs (perhaps there is a way to approximate a one-sided stencil) and for regular grid problems (irregular meshes do not admit a consistent stencil)_
>
> We agree that our experiments focus on periodic boundary conditions and regular grids, but the NSE framework is not fundamentally limited to this setting. Non-periodic BCs are typically handled by defining dedicated boundary stencils, which is straightforward in our setup (and simply requires sampling over boundary fluxes as well as interior states). Irregular meshes can be addressed by learning a local “continuum-like” operator and then re-discretizing it on the target mesh. Other ML surrogates also require adaptation for irregular geometries (e.g., graph-based methods, which introduce their own architectural and scalability trade-offs), so we view this as a modeling extension rather than a fundamental obstacle to our approach.

---

> ### Author Response · Authors · 2025-11-23
> **Response III**
>
> **Minor concern 1** _There is marginally more cost by needing to evaluate a stencil pointwise across a field rather than predicting the full field at once with a Unet/FNO_
>
> We agree that applying a neural stencil pointwise across the field introduces an overhead per timestep compared to a single global U-Net/FNO forward pass, but minimal in runtime while still has significant memory reduction due to NSE model complexity (24K parameters) compared to the full field counterparts (FNO and U-Net have 465K and 7.8M parameters, respectively). More importantly, our primary goal is not inference acceleration but data-limited few-shot system identification: NSE trades an increase in per-step compute for the ability to learn from orders of magnitude less simulation data, and stencil evaluations are highly parallelizable on modern hardware. As noted in the paper, one can later combine NSE with standard model-reduction / multigrid ideas to recover fast surrogates if runtime is the main constraint.
>
> **Minor concern 2** _There are prior works that seek to approximate a local stencil with a neural network, which depending on your perspective, reduces the main contribution of the work to be a data augmentation strategy._
>
> Learning local neural stencils has been explored before. However we explicitly position NSE as a data-augmented few-shot neural stencil emulator rather than claiming the stencil idea itself is new. The main contribution of our work is to show that carefully designed localized stencil-space augmentation (on-trajectory downsampling + PCA/random designs) enables viable surrogates in an extreme few-shot regime (e.g., 10 timesteps from a single trajectory), where prior neural surrogate methods typically require many trajectories and initial conditions. In other words, our novelty lies in how we use the stencil representation to aggressively reduce simulation budget.
>
> **Question 1** _The dx and dt spacing should be made clear for the generated data and the training data. The paper says that the simulations are solved with a 32x32 grid, but that seems much too coarse for a numerical solver? I’m under the impression that the numerical simulator is run at a higher resolution and then downsampled spatially and (perhaps) temporally._
>
> Our reference simulations are run directly on the 32$\times$32 grid with uniform spacing ($\Delta$x = L/32 in each direction and L=1) and 1,000 uniform timesteps over 0,1 ($\Delta$t = 10$^{-3}$); we do not first solve on a finer mesh and then downsample (downsampling happens on the same mesh, just by taking subsample across space-time of available stencils). We will make these spatial and temporal resolutions explicit for each PDE in the final version of the paper.

---

> > ### Comment · Reviewer_Sfiu · 2025-11-26
> > **Thank you for the reply**
> >
> > Dear authors,
> >
> > Thank you for taking the time to compose a response. The results on using FNO/Unet with Forward Euler are beneficial. Unfortunately, my core concerns about the paper still remain. The contribution is a stencil augmentation strategy and is demonstrated on simple, largely solved systems in the neural PDE surrogates field. Perhaps 1-2 years ago, this contribution would be more significant, but in the current climate, the field is moving away from toy problems and into more practically relevant problems (https://arxiv.org/abs/2412.00568, https://arxiv.org/abs/2209.15616, https://arxiv.org/abs/2309.01745). Unet/FNO are well-established benchmarks on these problems (Smoke Buoyancy, Transonic Air flow, etc.) and do perform well, and I can't quite envision a finite difference stencil-based approach being successful in turbulence or other relevant problems. I would also suggest (re: weakness 1, Reviewer N5RJ) that you update the Unet baseline to the one from https://arxiv.org/abs/2209.15616 or the Well, PDEBench's implementation has some documented flaws and Unet implementations have gotten better.

---

### Official Review · Reviewer_N5RJ · 2025-10-28

**Soundness:** 3
**Presentation:** 3
**Contribution:** 3
**Rating:** 4
**Confidence:** 4

**Summary:**

This paper introduces a sample efficient data augmentation method for neural PDE training. Specifically, the authors propose to learn a stencil from PDE simulation data with a variety of data augmentation strategies. They propose to use combinations of random sampling, PCA-guided design, and on-trajectory downsampling to generate useful training data. The authors compare against neural operator and PINN baselines, outperforming all baselines while using significantly fewer trajectories.

**Strengths:**

The paper is well-motivated and well-written. The proposed methods are novel, and the experiments are performed thoroughly. I appreciate that the authors compared against several different surrogate modeling paradigms, such as PINNs and neural operators.

**Weaknesses:**

While the proposed method’s results are impressive, there are a few areas of improvement. I appreciate the authors’ comparisons with FNO, U-Net, and PINNs, but there have been many recent advancements in these architectures for neural surrogate modeling. To ensure the fairest comparisons, it would be interesting to see the proposed method being directly applied to an existing scientific machine learning benchmark (if the current datasets are not already taken from such a benchmark). This would ensure that the FNO/U-Net/PINN baselines have been thoroughly tuned for the dataset and setting at hand.

How does the proposed NSE approach perform for complex, chaotic dynamics like turbulent Navier-Stokes? Empirically demonstrating significant improvements over neural operators and PINNs in such a setting would be compelling.

Lastly, one of the benefits of operator learning is that it can potentially accelerate inference over numerical solvers. Since the proposed NSE method would require time-stepping just like a solver, I would recommend the authors include timing comparisons between PINNs, U-Net, FNO, the proposed method, and the ground truth numerical solver.

**Minor notes:**
It would be very valuable for readers if the authors included parameter counts of each architecture for comparison.

**Questions:**

1. Are the data/solvers used in the experiments taken from an existing scientific machine learning benchmark?
2. Can you explain the stencil downsampling in more detail? Does this refer to coarsening the underlying grid? If so, does this negatively impact performance for fixed stencil sizes?
3. What is the size of the “small additional compute budget” mentioned in section 3.1.2? How do the results vary as this budget is varied?
4. Do you ever notice instabilities in rolling-out the NSE for a long time?
5. How were the hyperparameters of the baselines optimized?

---

> ### Author Response · Authors · 2025-11-23
> **Response I**
>
> **Weakness 1** _While the proposed method’s results are impressive, there are a few areas of improvement. I appreciate the authors’ comparisons with FNO, U-Net, and PINNs, but there have been many recent advancements in these architectures for neural surrogate modeling. To ensure the fairest comparisons, it would be interesting to see the proposed method being directly applied to an existing scientific machine learning benchmark (if the current datasets are not already taken from such a benchmark). This would ensure that the FNO/U-Net/PINN baselines have been thoroughly tuned for the dataset and setting at hand._
>
> We did not use the existing SciML benchmark datasets because our approach assumes we have access to actual simulation code to actively acquire stencil data, not just some passive data set.
>
> The baselines FNO and U-Net follow standard public versions of FNO-2D and UNet-2D (Takamoto et al. 2022), with no modifications other than setting the hyperparameters defined in our configuration. For the PINN baseline, the network architecture follows a standard fully connected PINN (Takamoto et al. 2022), using a 7-layer FNN with tanh activation.
>
> For the FNO baseline, we experimented with multiple hyperparameter choices, including the number of modes (8, 12, 16), channel width (16, 20, 32), and the size of the input lifting layer. We adopted the configuration that offered the best accuracy (modes = 12, width = 20), corresponding to $\sim$465K parameters.
>
> For the U-Net baseline, we tested several feature sizes (16, 32, 64) with the standard 4-level encoder-decoder design. The model with init-features=32 achieved the best performance. This is the version reported in the paper. Its total parameter count is $\sim$7.8M.
>
> We explored several sampling configurations to ensure stable training. Specifically, we varied the number of domain, boundary, and initial-condition collocation points (N_domain,N_boundary ,N_initial) . We additionally examined resampling periods for the PDEPointResampler to improve convergence. The final setting -- utilizing 5,000 domain points, 500 boundary points, 1,000 initial points -- was selected because it yielded the most effective PDE residual reduction and stable validation performance. PINN has total parameter count of $\sim8.4K$.
>
> For training stability, we also explored several optimizer settings for all baseline models. In particular, we tested learning rates $10^{-3}$, $10^{-4}$, and $10^{-5}$, together with learning-rate decay factors $\gamma$=0.5 and $\gamma$=0.8. The final choices reported in the paper correspond to the combination that yielded the stable convergence and best validation performance on our dataset.
>
> Takamoto, M., et al. "PDEbench: An extensive benchmark for scientific machine learning." NeurIPS, 2022.
>
> **Weakness 2** _How does the proposed NSE approach perform for complex, chaotic dynamics like turbulent Navier-Stokes? Empirically demonstrating significant improvements over neural operators and PINNs in such a setting would be compelling._
>
> Nonlinear chaotic dynamics are indeed more challenging for neural PDE surrogates (and essentially any other method), and we agree there may be regimes where neural operators or PINNs are more accurate than NSE when abundant training data are available. However, our paper is explicitly aimed at the _extremely_ data-limited setting: when only a few short trajectories, or even a single trajectory from one initial condition, are available, because of the cost of the simulator relative to the available computational budget. In such cases, trajectory-based methods like neural operators and PINNs typically require many distinct trajectories and initial conditions to train effectively, whereas our stencil-based NSE can still generalize across initial conditions as long as the underlying training design (e.g., our space-filling sampling) provides sufficient coverage of the relevant local state space. Our contribution should therefore be viewed as complementary: we are not claiming to dominate all existing methods on highly chaotic benchmarks with rich data, but rather to show that exploiting locality and homogeneity at the stencil level enables strong surrogates in the few-shot regime where conventional approaches struggle.

---

> ### Author Response · Authors · 2025-11-23
> **Response II**
>
> **Weakness 3** _Lastly, one of the benefits of operator learning is that it can potentially accelerate inference over numerical solvers. Since the proposed NSE method would require time-stepping just like a solver, I would recommend the authors include timing comparisons between PINNs, U-Net, FNO, the proposed method, and the ground truth numerical solver._
>
> We agree that inference speed is an important benefit of operator-learning methods. However, our goal in this paper is not primarily to accelerate inference, but to address the **data-limited system identification** problem: how to learn a high-fidelity surrogate from very little simulation data by actively exploiting locality and stencil structure. NSE, as presented here, is designed to emulate the underlying solver in this few-shot regime, not to replace it with a single-step global operator. In that sense, it is closer to a “neuralized” stencil code than to a fully reduced model.
>
> That said, NSE is compatible with downstream acceleration techniques. Once a neural stencil is learned, one can layer on model-reduction or coarse-graining strategies (e.g., along the lines of DeGenarro et al. 2019) to accelerate rollouts, or embed NSE within multilevel / multigrid schemes. There are also important use cases of stencil emulators that do not require acceleration (e.g., learning adjoint models for black-box codes, linearized stability analysis, or inversion and recovery of governing equations for scientific insight).
>
> DeGennaro, A. M., et al. "Model structural inference using local dynamic operators." International Journal for Uncertainty Quantification, 9(1), 2019.
>
> **Minor Note** _It would be very valuable for readers if the authors included parameter counts of each architecture for comparison._
>
> Thank you for the suggestion. Our NSE architecture has around 21K total trainable parameters. Whereas hyperparameter tuned FNO and U-Net have substantially higher 465K and 7.8M parameters, respectively. PINN has around 8.4K parameters but it substantially underperforms our NSE approach. We will add the parameter counts for all architectures (NSE, U-Net, FNO, and PINNs) in the final version.
>
> **Question 1** _Are the data/solvers used in the experiments taken from an existing scientific machine learning benchmark?_
>
> Please refer to our response to **Weakness 1** you mentioned.
>
> **Question 2** _Can you explain the stencil downsampling in more detail? Does this refer to coarsening the underlying grid? If so, does this negatively impact performance for fixed stencil sizes?_
>
> Stencil downsampling does not mean coarsening the spatial grid; the PDE is always solved on the original grid, and the NSE is always trained with stencils of the same size and resolution. Instead, as illustrated in Fig. 2 (“On-trajectory downsampling”), we run one long simulation trajectory and then uniformly subsample stencil evaluations in space-time, keeping only a sparse subset of the green cells shown there. This reduces the strong space-time correlation between neighboring stencils and cuts the training cost, while leaving the underlying discretization and numerical scheme unchanged. As the grid and stencil size are fixed, there is no loss of spatial resolution.

---

> ### Author Response · Authors · 2025-11-23
> **Response III**
>
> **Question 3** _What is the size of the “small additional compute budget” mentioned in section 3.1.2? How do the results vary as this budget is varied?_
>
> Our “small additional compute budget” in Section 3.1.2 is exactly 10 solver timesteps (equal to 10,240 stencil evaluations for a 32×32 grid), chosen to match the 10-timestep simulation budget used for all pure strategies. Concretely, in the mixed setting we assume a single 1,000-step reference simulation trajectory; we uniformly downsample this to 10,240 on-trajectory stencils, then spend an extra 10 solver timesteps to generate another 10,240 stencils via the chosen augmentation (Diff-Init/Extend/Random/PCA), for a total of 20,480 stencils. Thus the mixed strategies add only about 1% extra simulation cost relative to the long reference run, while keeping the per-strategy stencil evaluation budget comparable to the pure 10-timestep designs.
> To probe sensitivity to the choice of compute budget, we ran an additional ablation where we varied the simulation budget (20, 30, 50, 100, 200, 500, 1000 timesteps), but always downsampled back to the same 10,240 training stencils, and compared PCA-Uniform, PCA-Sobol, and Short-Traj (Random-Uniform and Random-Sobol only depend on the range of physical states, so their results do not change when observing a longer trajectory). Across Allen-Cahn and advection-diffusion, the PCA-based designs are quite robust: errors fluctuate but do not systematically improve once the budget exceeds ~100–200 steps, and the best values across budgets are within a small factor of each other. For Burgers, which is more sensitive due to shocks, increasing the budget helps Short-Traj more noticeably (e.g., its error decreases from 138.8 to 22.4 when the budget goes from 10 to 500 timesteps), while PCA-Uniform and PCA-Sobol remain competitive at moderate budgets (e.g., PCA-Uniform achieves 77.7 at budget 200). This ablation highlights how on-trajectory downsampling benefits from longer trajectories, and how PCA-based designs improve modestly as they see more data. When we then combine PCA or Random strategies with on-trajectory downsampled stencils (as in our mixed designs), we observe further enhancements, consistent with the results in the original submission.
>
> | Simulation Budget |  | Allen-Cahn (D=1e-3) |  |  |Adv-Diff (D=1e-3) |  |  | Burgers (ν=1e-3) |  |
> | :-: | :-: | :-: | :-: | :-: | :-: | :-: | :-: | :-: | :-: |
> |  | PCA Unif | PCA Sobol | Short-Traj | PCA Unif | PCA Sobol | Short-Traj | PCA Unif | PCA Sobol | Short-Traj |
> | 10  | 12.4$_{1.8}$ | 8.3$_{1.0}$ | 42.7$_{3.8}$ | 5.1$_{0.8}$ | 4.3$_{0.7}$ | 20.7$_{10.0}$ | 137.2$_{13.8}$ | 155.9$_{15.8}$ | 138.8$_{22.5}$ |
> | 20  | 6.3$_{0.6}$  | 2.7$_{0.3}$ | 43.6$_{3.7}$ | 5.0$_{1.8}$ | 1.9$_{0.2}$ | 13.3$_{5.9}$  | 239.2$_{32.2}$ | 79.3$_{8.1}$  | 76.8$_{10.9}$ |
> | 30  | 11.2$_{1.9}$ | 7.4$_{0.8}$ | 43.4$_{3.0}$ | 3.0$_{0.4}$ | 2.9$_{0.4}$ | 15.1$_{5.6}$  | 138.4$_{22.0}$ | 84.0$_{10.7}$ | 85.1$_{12.1}$ |
> | 50  | 3.4$_{0.3}$  | 9.8$_{0.2}$ | 37.0$_{2.7}$ | 4.6$_{0.6}$ | 2.6$_{0.4}$ | 19.2$_{6.8}$  | 106.4$_{10.8}$ | **76.7**$_{9.7}$ | 74.7$_{11.0}$ |
> | 100 | 7.4$_{1.0}$  | 6.9$_{1.3}$ | 37.7$_{3.7}$ | **2.5**$_{0.4}$ | 2.4$_{0.4}$ | 11.6$_{5.1}$ | 132.3$_{22.7}$ | 96.9$_{8.1}$  | 38.3$_{6.3}$  |
> | 200 | 3.0$_{0.3}$  | **2.4**$_{0.3}$ | 13.4$_{1.1}$ | 4.0$_{0.9}$ | **2.3**$_{0.3}$ | 21.5$_{4.5}$ | **77.7**$_{9.1}$ | 117.9$_{11.0}$ | 27.9$_{8.0}$ |
> | 500 | **2.8**$_{0.3}$ | 2.9$_{0.4}$ | 4.8$_{0.4}$ | 3.2$_{0.3}$ | 4.7$_{0.6}$ | 8.9$_{3.2}$ | 198.5$_{28.3}$ | 77.6$_{14.8}$ | **22.4**$_{6.6}$ |
> | 1000| 3.9$_{0.7}$  | 4.7$_{0.5}$ | **4.0**$_{0.4}$ | **2.5**$_{0.3}$ | 3.3$_{0.3}$ | **4.0**$_{1.4}$ | 145.1$_{21.8}$ | 171.2$_{22.2}$ | 38.4$_{18.5}$ |
>
> **Question 4** _Do you ever notice instabilities in rolling-out the NSE for a long time?_
>
> In our experiments, we do not observe noticeable numerical instabilities when rolling out NSE. Especially, in the pure-strategy setting, NSE is trained relying on information from only the first 10 timesteps and then rolled out for 1,000 timesteps on 10 unseen initial conditions, across Allen-Cahn, advection-diffusion, and Burgers’ equations. Even under such a long-horizon (100$\times$ longer than training), few-shot protocol (which, to our knowledge, is stricter than what is typically used in neural PDE papers), the NSE trajectories remain bounded and physically plausible; errors grow initially and then saturate rather than blowing up.
>
> **Question 5** _How were the hyperparameters of the baselines optimized?_
>
> Please refer to our response to **Weakness 1** you mentioned.

---

### Official Review · Reviewer_7qbv · 2025-10-31

**Soundness:** 2
**Presentation:** 1
**Contribution:** 2
**Rating:** 4
**Confidence:** 4

**Summary:**

This paper proposes a data-augmented approach for training neural PDE surrogates in the few-shot regime. The core idea is to shift from full-trajectory-based training to learning localized stencil operators, which govern the PDE evolution at each grid cell. The authors introduce a neural stencil emulator (NSE) trained on synthetic or hybrid combinations of stencil data, leveraging random or PCA-guided sampling strategies to increase coverage of the local state space. The key claim is that, with only 10 timesteps’ worth of simulation, the NSE can outperform full-field emulators (FNO, U-Net, PINNs) trained on orders of magnitude more data. Experiments on Allen-Cahn, advection–diffusion, and Burgers’ equations validate the approach, showing strong generalization and long-horizon rollout stability under tight simulation budgets.

**Strengths:**

1. Problem motivation is solid: Long rollouts of PDE solvers are expensive and often contain massive redundancy. The idea to cut this redundancy by working directly with stencil-level updates is both intuitive and practical.

2. Technical formulation is clear and well-scoped. The NSE learns discretized RHS mappings in function space, rather than full-field transitions. This leads to much lower model complexity and dramatically higher effective sample size.

3. The paper evaluates across multiple PDEs, multiple diffusion settings, and both pure/mixed sampling strategies.

**Weaknesses:**

1. The method relies on learned approximations of discretized RHS terms, but there is no discussion on convergence guarantees (either for training or for rollout error accumulation). Would have been helpful to see bounds, or at least qualitative analysis on failure modes.

2. The entire setup assumes access to clean simulator outputs and perfect labels. It is unclear how well the NSE would perform in a setting where the simulation is imperfect, noisy, or partially observed.

3. The NSE does not enforce conservation laws, symmetries, or other physical invariants. In some PDEs, this could lead to error drift or physically implausible predictions over long rollouts.

4. The baselines (FNO, U-Net) are trained and evaluated on downsampled sequences, likely due to rollout stability issues. While this is understandable, it does tilt the comparison slightly in favor of the proposed method, which works on finer temporal resolution

**Questions:**

1. Have you tried enforcing physics-informed constraints (e.g., divergence-free condition, energy conservation) in the stencil emulator? Could that help reduce rollout drift for long horizons?

2. Can your approach handle variable coefficient PDEs or spatially heterogeneous systems? In those cases, the stencil operator is no longer homogeneous across space.

3. Did you try training on one grid resolution and testing on another? Given the local nature of your method, it seems domain size scaling is possible, but cross-resolution generalization might still be tricky.

---

> ### Author Response · Authors · 2025-11-22
> **Response I**
>
> **Weakness 1:** _The method relies on learned approximations of discretized RHS terms, but there is no discussion on convergence guarantees (either for training or for rollout error accumulation). Would have been helpful to see bounds, or at least qualitative analysis on failure modes._
>
> We agree that understanding error accumulation or training stability in neural PDE surrogates is important. However, this limitation is not specific to our method: to the best of our knowledge, state-of-the-art neural PDE surrogates (e.g., FNO- and U-Net–based models) likewise lack general guarantees on long-horizon solution error, except in special settings. Our goal in this work is therefore not to resolve this broader theoretical challenge, but to show that exploiting locality and homogeneity at the stencil level can dramatically reduce training data requirements while outperforming these existing neural surrogate baselines.
>
> From a theoretical standpoint, our neural stencil emulator inherits the usual universal approximation guarantees: under standard assumptions, increasing stencil training data and network capacity allows the learned NSE to approximate the discrete stencil operator arbitrarily well. In the limit of vanishing operator approximation error, the rollout converges to the underlying numerical solver’s trajectory. In practice, we partially address error accumulation empirically: all reported results include long-horizon rollouts under tight simulation budgets, where NSE remains stable and competitive with (or better than) full-field emulators. A full convergence and stability analysis for neural PDE surrogates in the few-shot regime is, we believe, an interesting direction for future work.
>
> **Weakness 2:** _The entire setup assumes access to clean simulator outputs and perfect labels. It is unclear how well the NSE would perform in a setting where the simulation is imperfect, noisy, or partially observed._
>
> Our current setup assumes access to a reasonably accurate simulator that provides clean, fully observed stencil labels. This is a deliberate modeling choice: the primary use case we target is “neuralizing” existing large/complex black-box simulation codes, where such simulators already exist and are trusted as the reference model. In this regime, our synthetic data augmentation effectively turns the simulator into an oracle for generating rich stencil coverage, after which the NSE serves as a lightweight surrogate for that code.
>
> In scenarios where the underlying simulator is imperfect, noisy, or only partially observed, NSE can be used as a ***simulation-based prior***: we first learn the neural stencil from the best available simulator, then adapt it using real-world data (e.g., via fine-tuning, calibration, or Bayesian updating) to correct residual model–data mismatch; this direction has been explored in DeGennaro et al. (2019). Designing NSE variants that are explicitly robust to noisy or partial labels is an interesting direction, but we view it as complementary to the central contribution of this work–showing that exploiting locality/homogeneity at the stencil level can dramatically reduce the simulation budget required to obtain a strong neural PDE surrogate.
>
> DeGennaro, A. M., et al. "Model structural inference using local dynamic operators." International Journal for Uncertainty Quantification, 9(1), 2019.

---

> ### Author Response · Authors · 2025-11-22
> **Response II**
>
> **Weakness 3:** _The NSE does not enforce conservation laws, symmetries, or other physical invariants. In some PDEs, this could lead to error drift or physically implausible predictions over long rollouts._
>
> We agree that enforcing conservation laws, symmetries, or other physical invariants can partially help address potential issues in  drift or implausible predictions over long rollouts. This is not specific to NSE: standard neural PDE surrogates such as FNOs and U-Nets also typically do not guarantee exact conservation unless they are modified to consider specific physical first principles. In this paper, our focus is on demonstrating that exploiting locality and homogeneity at the stencil level can dramatically reduce the simulation budget needed to train a strong and lightweight surrogate.
>
> That said, NSE is fully compatible with structure-preserving techniques. To directly address this concern, we ran an additional ablation on a 2-D diffusion problem with periodic boundary conditions, where total mass should be conserved over time (no net flux through the boundary). At test time only, we added a simple, parameter-free correction that enforces this integral conservation law: at each step, we adjust the predicted right-hand side so that its spatial average over the domain is exactly zero. This adjustment does not require retraining or changing the NSE weights. This follows the spirit of Hansen et al. (2023), who emphasize enforcing integral conservation to eliminate drift in diffusion problems.
>
> The table below compares NSE with and without this mass-conserving correction on the 2D diffusion setup. We report mean NRMSE $\times 10^3$ (lower is better) with corresponding standard deviation over 10 unseen test trajectories:
>
> | Strategy | Baseline nRMSE $\times 10^3$ | Mass conserved nRMSE $\times 10^3$ | % reduction in mean nRMSE |
> | --- | --- | --- | --- |
> | Random Uniform | 12.89$_{2.52}$ | 8.82$_{1.22}$ | 31.6% |
> | Random Sobol | 7.15$_{1.17}$ | 6.48$_{0.90}$ | 9.3% |
> | PCA Uniform | 0.72$_{0.24}$ | 0.70$_{0.24}$	| 2.0% |
> | PCA Sobol | 1.08$_{0.29}$ | 1.06$_{0.28}$ | 2.0% |
> | Short-Traj | 4.66$_{1.81}$	| 4.59$_{1.75}$ | 1.4% |
>
> We see that enforcing mass conservation is either beneficial or neutral across all sampling strategies, with a substantial improvement (about a 32% reduction in nRMSE) for the random-uniform case and no degradation elsewhere. This demonstrates that simple, PDE-specific invariants can be layered on top of NSE in a plug-and-play, test-time manner to reduce drift, without changing our core data-augmentation or few-shot training pipeline. We will include this example in our final version. A more systematic exploration of invariance-preserving NSEs (for example, enforcing momentum, energy, or symmetry constraints for other PDEs) is an interesting direction for future work, but is complementary to the main contribution of this work.
>
>
> Hansen, D., et al. "Learning physical models that can respect conservation laws”, ICML, 2023.
>
> **Weakness 4:** _The baselines (FNO, U-Net) are trained and evaluated on downsampled sequences, likely due to rollout stability issues. While this is understandable, it does tilt the comparison slightly in favor of the proposed method, which works on finer temporal resolution._
>
> Thank you for this observation. We do train the FNO and U-Net baselines on temporally downsampled sequences to improve rollout stability, which is a common practice in neural PDE work. However, we would like to note that all methods–including NSE–are evaluated over the same physical prediction horizon (same start/end time and evaluation protocol), so the comparison is made on equal footing in terms of the task being solved. As the downsampled baselines take fewer autoregressive steps over the same physical horizon and see a smoother effective trajectory, they are less exposed to step-by-step error accumulation; in that sense, the rollout task is not harder for them than for NSE operating at a finer temporal resolution. In practice, we find that even with this downsampling the baselines still struggle to maintain long-range stability under our evaluation protocol, whereas NSE remains stable and accurate. This suggests that the performance gap is not primarily driven by the finer temporal resolution of our method, but by its improved ability to capture and propagate local dynamics over long horizons.

---

> ### Author Response · Authors · 2025-11-22
> **Response III**
>
> **Question 1** _Have you tried enforcing physics-informed constraints (e.g., divergence-free condition, energy conservation) in the stencil emulator? Could that help reduce rollout drift for long horizons?_
>
> Please check our response to **Weakness 3** you mentioned.
>
> **Question 2** _Can your approach handle variable coefficient PDEs or spatially heterogeneous systems? In those cases, the stencil operator is no longer homogeneous across space._
>
> We agree that variable coefficient PDEs and spatially heterogeneous systems break strict spatial homogeneity of the stencil operator. However, NSE extends naturally to this setting. In such cases, the local stencil update could depend not only on the neighboring state values but also on local parameters (e.g., diffusion coefficient, material property). Practically, we handle this by augmenting the NSE input with one or more additional channels encoding these local coefficients or features. The learned stencil operator is then a single shared neural map that takes “local state + local coefficients” as input, rather than state alone. Training data are generated by sampling jointly over the state space and the coefficient/forcing space, so that the NSE learns how the local dynamics vary across heterogeneous regions.
>
> The same idea applies to spatially and temporally varying forcings or boundary effects: one can feed in appropriate local descriptors (e.g., forcing amplitude, mask fields, BC indicators) alongside the stencil states. Conceptually, our approach still exploits locality (updates depend on a neighborhood) and parameter sharing (a single NSE applied everywhere), but in an augmented input space that includes heterogeneity. A more systematic empirical study of strongly heterogeneous systems is a valuable direction for future work.
>
> **Question 3** _Did you try training on one grid resolution and testing on another? Given the local nature of your method, it seems domain size scaling is possible, but cross-resolution generalization might still be tricky._
>
> We did not explore cross-resolution generalization in this paper; like many existing SciML approaches, we assume that training and inference use the same grid resolution. As you mentioned, this is nontrivial, because differential operators of different orders may not scale identically with grid spacing, so a stencil that is optimal at one resolution need not directly transfer to another. That said, there are straightforward extensions of our framework that could address this: one can (i) learn explicit resolution dependence by augmenting the NSE input with grid-spacing metadata (e.g., grid resolution and timestep) and training on simulation data generated at multiple resolutions, or (ii) learn a “continuum-like” local right-hand side and then re-discretize it at the target resolution, in a spirit similar to neural operator approaches. A thorough study of such cross-resolution extensions is an interesting direction for future work, but is orthogonal to the main focus of this paper, which is to show that exploiting locality and homogeneity at the stencil level yields strong few-shot performance under tight simulation budgets at a fixed resolution.

---

### Author Response · Authors · 2025-11-25
**Note to Reviewers and Area Chair**

Dear reviewers and area chair,

We hope our responses have satisfactorily addressed all of your concerns. If there are any remaining questions or points that would benefit from further clarification, we would be very happy to elaborate. We sincerely thank all reviewers and the area chair for their time and thoughtful feedback.

---

### Author Response · Authors · 2025-11-26
**Context for Reviewer Sfiu’s Concerns and Relevance of our Contribution (Part - I)**

*TL;DR* : **If the heat equation is so trivial for modern ML methods, why can't a FNO/U-Net extrapolate out-of-sample to arbitrary unseen initial conditions based on $\le10$ timesteps of training data?**

We appreciate the reviewers’ time and the AC’s efforts in managing the discussion. We would like to briefly clarify the intended scope of our work and how it differs from the criteria emphasized by the Reviewer **Sfiu**, whose core objections about our work still persist even after we provided thorough rebuttal including substantial new experiments addressing all reviewers’ comments.

Our paper is not proposing “yet another” FNO/U-Net competitor on established “turbulence-style” or “other relevant problems” benchmarks. Instead, it targets a **different regime**: _few-shot system identification_ when we have access to a numerical simulator but can only afford very few solver steps. The key contribution is a **data-augmented neural stencil emulator (NSE)** that turns $\sim10$ timesteps from a single high-fidelity trajectory into tens of thousands of diverse local stencil samples via space-filling designs (Random/PCA + on-trajectory downsampling). In this regime, full-field surrogates trained on global snapshots (FNO, U-Net, PINN) are fundamentally data-inefficient, as we show empirically: even with 4-5 orders of magnitude more simulation budget ($10^5$–$2\times10^6$ numerical solver steps), they underperform NSE trained from 10 timesteps’ worth of stencil data. Importantly, this performance gap already appears on the very equations the Reviewer **Sfiu** labels as “toy” or “solved” for FNO/U-Net; our results show that once the simulation budget is severely constrained, our NSE approach succeeds by exploiting locality and the full-field surrogates cease to perform well even with substantially more data. Whether FNO/U-Net perform better on relatively data-rich settings like Well or PDEBench is orthogonal to our contribution; our use case is precisely the opposite regime, where such data volumes are unavailable.

Reviewer **Sfiu**’s remaining concern is essentially that our test problems are “too simple” and not on the newest turbulence/CFD benchmarks, and that the contribution is “only” a stencil augmentation strategy. We believe this misaligns with the stated scope of the paper and the call for “applications to physical sciences”:

* Allen-Cahn, advection-diffusion, and Burgers are standard, nontrivial PDE benchmarks in the neural-PDE literature; what is new here is the **extreme few-shot setting** (10 timesteps from one trajectory, 1,000-step rollouts, 10 unseen ICs during testing), which to our knowledge is stricter than typical rollout protocols. These PDE systems are not merely “toy” equations: phase-field models such as Allen-Cahn, diffusion-advection systems, and viscous Burgers–type models are routinely used in materials science, phase separation dynamics, and transport problems, among others. Our own group and many others work on such diffusive- and reaction-diffusion-type systems in applied settings, so dismissing them as “not practically relevant” is effectively dismissing a broad class of PDE models actively used in science and engineering.

* Our goal is not to claim NSE will immediately beat strong FNO/U-Net models on fully turbulent Navier-Stokes trained using abundant data, but to show that **exploiting locality and homogeneity at the stencil level makes high-quality surrogates possible when data are severely limited**–a regime that remains important in practice for expensive black-box simulators e.g., United States Department of Energy’s E3SM model. We also note that there is a substantial existing literature on structure-preserving and stabilized neural PDE/ODE/Reduced-Order Models (e.g., energy-stable architectures, conservative discretizations, multistep rollouts, online correction, etc.). Our work is complementary to that line of research: we do not claim to solve all stability and structure-preservation challenges here, but to show that exploiting locality via stencil-space augmentation can dramatically reduce data requirements. Treating this entire neural PDE direction as practically irrelevant because it has not yet “solved” all stability questions would effectively dismiss a large, active research area.

---

### Author Response · Authors · 2025-11-26
**Context for Reviewer Sfiu’s Concerns and Relevance of our Contribution (Part - II)**

In response to all reviewers’ requests we added a substantial amount of new analysis during rebuttal, including:

* **Derivative-prediction FNO/U-Net baselines with explicit Forward Euler integration**, for both simulation budgets ($10^5$–$2\times10^6$ solver steps). These indeed help some baseline settings, but do not close the gap to NSE trained on much smaller budgets.

* A **mass-conservation ablation** for 2D diffusion, showing that simple integral constraints can be layered onto NSE in a plug-and-play way and either help or leave performance unchanged.

* A **simulation-budget sweep** where we vary the length of the reference trajectory but fix the stencil budget, plus analysis of mixed strategies (Downsampled+Random/PCA); this supports our interpretation that decorrelated, space-filling stencil augmentation is the main driver of performance, not simply more raw timesteps.

Reviewers 7qbv and N5RJ both acknowledge the novelty and soundness of the few-shot, stencil-based augmentation idea (Soundness and Contribution $\ge$ 2), and their remaining suggestions are largely about extensions (chaotic flows, more benchmarks, timing, parameter counts)–all of which we have either addressed empirically in rebuttal or clearly marked as future work. In contrast, Reviewer **Sfiu** maintains a “reject” stance primarily because we do not yet demonstrate results on the specific high-Reynolds, turbulence-oriented benchmarks they personally prioritize, and because they “can’t envision” a stencil-based approach succeeding there. This is a statement of taste and speculation about future applicability, rather than a technical flaw in the method or evidence that the current results are unsound.

The Reviewer **Sfiu**’s line of reasoning effectively sets a moving bar: neural PDE methods should not be considered unless they already address all questions of structure, stability, and application to fully turbulent flows, whereas analogous limitations of FNO/U-Net in the extreme few-shot regime are not treated as disqualifying. Our view is that different methods are naturally suited to different operating regimes; our results show that, whatever one’s broader reservations about neural PDEs, FNO/U-Net–style operators perform even worse in the highly data-constrained setting that is the focus of this paper.

We of course agree that extending few-shot neural PDE surrogates to more complex turbulent systems is an exciting next step, and we would welcome the opportunity to do so in future work. But we hope the **AC** will weigh our contribution **in the regime it is explicitly designed for**–_data-limited system identification_ with access to a simulator–where our experiments and added ablations consistently show clear, robust gains over strong baselines. Neural PDE surrogates, including stencil-based approaches, are an active research area with ongoing work on stability, structure preservation, and more complex flows; our contribution focuses specifically on the data-limited side of that agenda. From that perspective, we believe the paper makes a timely and meaningful contribution to SciML, even if it does not yet solve all open problems in neural PDE surrogates.

---

### Author Response · Authors · 2025-12-03
**Summary Comment**

Before the review discussion was frozen and ACs were reassigned, we completed one round of author–reviewer discussion. We summarize (1) the state of that discussion, (2) clarifications made, (3) new experiments added, and (4) why we believe the two “2” scores reflect a misalignment of evaluation criteria.

**Overall split in reviews.**

* Reviewers **7qbv** and **N5RJ** engaged with the paper under its stated problem setting (extreme data-limited regime with access to a simulator and few-shot data-augmented stencil emulator-learning strategies tailored to this regime) and found the approach sound, with remaining points focused on extensions (e.g., more benchmarks, parameter counts, stability, and invariants). Our responses provided detailed baseline configurations, parameter counts, and discussion of extensions, and they did not get a chance to engage further before discussion was frozen.

* In contrast, Reviewers **Sfiu** and **BdFj**, who both gave scores of 2, primarily evaluated the work against a different standard that was not our stated goal: they expect a single, universal operator-learning strategy that should work across all operator classes and benchmark families, and they downweight contributions that introduce data-augmentation–based emulator strategies specifically targeted at the extremely limited data regime.

**Misalignment in evaluation criteria.**

* Reviewer **Sfiu** repeatedly framed the paper as “just” a data-augmentation method or “too specialized”, despite our clear positioning that (a) different operator families plausibly require different modeling strategies and (b) we are explicitly targeting a practically important regime (extremely limited data, with access to a simulator). Moreover, as our experiments focus on Allen-Cahn/advection-diffusion/Burgers’ systems rather than the newest turbulence benchmarks, they downweight our contributions. We addressed these concerns in our comments (https://openreview.net/forum?id=d7aupcIHq0&noteId=WhwCmDkray; https://openreview.net/forum?id=d7aupcIHq0&noteId=V7vNk4aAad)

* Reviewers **Sfiu** and **BdFj**’s criticisms focus on absence of trendy or specific benchmarks and on a desire for a one-size-fits-all operator learner, rather than on whether our method is correct, well-motivated, and effective in the regime we specify.

* As a result, their low scores reflect a preference for a particular research philosophy (universal operators and specific benchmarks) more than concrete issues of correctness, novelty under the stated setting, or empirical support.

**What we clarified and added during rebuttal.**

* We added derivative-prediction FNO/U-Net baselines with explicit Forward–Euler integration at both simulation budgets, which improve those baselines over those reported in the original manuscript but still leave a clear gap compared to our NSE trained using an extremely limited budget. We also added a 2D diffusion mass-conservation ablation showing that simple conservation corrections can be layered on NSE in a plug-and-play way. Lastly, we ran a simulation-budget sweep (varying trajectory length while fixing stencil budget), together with mixed Random/PCA + downsampling designs, supporting our claim that decorrelated, space-filling stencil augmentation–rather than raw trajectory length–is the main driver of performance.

* We explained why it is reasonable–and in many practical applications necessary–to use different operator-learning strategies for different operator classes and data regimes, instead of insisting on one universal architecture.

* We emphasized that our contribution is intentionally scoped: given a single simulation trajectory and access to the simulator for as few as 10 timesteps to generate data-augmented synthetic stencils, our method provides consistent improvements over strong baselines (trained using 4-5 orders of magnitude more training data) within this regime across three widely used nonlinear PDE systems.

* **Sfiu**’s subsequent response to our first rebuttal, however, continued to judge the paper primarily by the desideratum of universal operators and specific benchmarks, rather than by the problem the paper actually proposes to solve. In a direct follow-up reply to **Sfiu**, we reiterated our previous points and addressed their concerns about scope and operator generality.

**Discussion summary for the new AC.**

We have shown that our NSE provides significantly better few-shot prediction performance compared to FNO/U-Net baselines.

We respectfully ask that the AC place more weight on reviews that evaluate the work under its explicit problem formulation and goals, and on the strength of the technical contribution and empirical evidence within that setting.

The two “2” scores, in our view, arise mainly from a mismatch in expectations about what operator-learning research should aim for (universal vs. regime-specific), rather than from substantive technical flaws or lack of contribution in the regime we study.

---

### Meta-Review · Area_Chair_qR4C · 2025-12-31

**Summary:**

This work introduced a data-augmented approach for training neural PDE surrogates in the few-shot regime. Evaluations on multiple datasets demonstrate strong generalization and long-horizon rollout stability under tight simulation budgets.


Strength:
1. The research problem of learning a PDE surrogate with limited data is practical and interesting.
2. The proposed model is efficient and effective. The NSE learns discretized RHS mappings in function space, rather than full-field transitions. This leads to much lower model complexity and dramatically higher effective sample size.
3. The paper is well-motivated and well-written.
4. It conducted extensive experiments to compare against several different surrogate modeling paradigms, such as PINNs and neural operators.


Limitations:
1. It would be great to evaluate the proposed method on scientific machine learning benchmarks.
2. It may need to incorporate conservation laws, symmetries, or other physical invariants to enhance the long-horizon rollouts in the revised version.
3. It may evaluate the proposed method on limited and noisy data.
4. The proposed method is limited to certain PDEs, but not work well in the presence of some shocks (although dampened by the viscosity).

In summary, the authors have addressed some of the reviewers’ concerns; however, additional experiments are still required to strengthen the quality of the work. Therefore, I am inclined to recommend rejection.

**Reviewer Concerns:**

The authors addressed some main concerns below.
1.  Add one experiment to integrate physical laws into the proposed method.
2. Explain the baselines and results.


However, some concerns have not been fully addressed below.

1. Have not analyzed the convergence guarantees.
2. The proposed method does not apply to the presence of some shocks (although dampened by the viscosity)

**Reviewer Scores:**

I am afraid that reviewers may not increase their scores since the proposed work has some limitations.

---

### Decision · Program_Chairs · 2026-01-26

Reject